# Seasonal pigment fluctuation in diploid and polyploid *Arabidopsis* revealed by machine learning-based phenotyping method PlantServation

Reiko Akiyama [1], Takao Goto[2], Toshiaki Tameshige [3,4], Jiro Sugisaka[3,5], Ken Kuroki [6], Jianqiang Sun [7], Junichi Akita[8], Masaomi Hatakeyama[1,9], Hiroshi Kudoh [5], Tanaka Kenta[10], Aya Tonouchi[2], Yuki Shimahara[2], Jun Sese [11,12,13], Natsumaro Kutsuna[2], Rie Shimizu-Inatsugi [1] ✉ & Kentaro K. Shimizu [1,3] ✉

Long-term field monitoring of leaf pigment content is informative for understanding plant responses to environments distinct from regulated chambers but is impractical by conventional destructive measurements. We developed PlantServation, a method incorporating robust image-acquisition hardware and deep learning-based software that extracts leaf color by detecting plant individuals automatically. As a case study, we applied PlantServation to examine environmental and genotypic effects on the pigment anthocyanin content estimated from leaf color. We processed >4 million images of small individuals of four *Arabidopsis* species in the field, where the plant shape, color, and background vary over months. Past radiation, coldness, and precipitation significantly affected the anthocyanin content. The synthetic allopolyploid *A. kamchatica* recapitulated the fluctuations of natural polyploids by integrating diploid responses. The data support a long-standing hypothesis stating that allopolyploids can inherit and combine the traits of progenitors. PlantServation facilitates the study of plant responses to complex environments termed "*in natura*".

Plants in field environments (hereafter referred to as "*in natura*") are exposed to complex environments with multiple abiotic and biotic factors[1]. Therefore, knowledge from indoor studies, mostly focusing on a single factor, is not necessarily directly transferrable to the field[2–4]. Time-series data from the field are informative for understanding how plants thrive. However, collecting time-series data on plant growth and response in fluctuating environments is labor-intensive and often destructive[5]. Image analysis serves as a non-destructive alternative for time-series data collection in the field; however, the acquisition and analysis of high-resolution time-series images from the field is challenging for several reasons. First, a setup is exposed for months to various weather conditions, such as sunlight, rain, snow, or storms. Such robust systems are often costly[5]. Second, even with fixed-point image acquisition, the camera positions change owing to extreme weather conditions, maintenance work, etc., resulting in the inconsistency of the position of target plants in images from different time points. Third, plant segmentation in acquired images may not be straightforward[6,7]. Plant size, morphology, and color can change over time or are affected by conditions such as snowfall. External conditions such as light intensity, soil texture, and wind disturbance also vary among sites, introducing variations among images from different sites. Fourth, the ideal image resolution to observe a small single individual,

such as seedlings of model *Arabidopsis* species, may be difficult to achieve with the currently available cost-efficient methodologies. The resolution of close-up images by drones is typically of the order of centimeters or at best millimeters because of the disturbance of plants caused by the wind that the drones generate[5,8]. In contrast to drones, ground-based systems can easily capture close-up images; however, commercially available products are intended for large-scale fields and are not cost-efficient for small-scale research[9]. Yang et al. (2022) and Hawkesford and Lorence (2017) emphasized that it is important to decrease the cost of phenotyping to promote further research[5,10]. Finally, the amount of collected data can be large, resulting in a long processing time for analysis. Overcoming all these challenges and analyzing time-series images of different species in different environments further our understanding of the growth and environmental responses of plants.

Deep neural network (DNN) is a powerful tool for analyzing complex images. DNN is increasingly being deployed to process large image datasets in diverse disciplines, from medical science to engineering[11]. In plant science, it has been successfully implemented to segment the target plant or the position of the plant in an image under controlled conditions, where the plant appearance and background are relatively uniform[12,13]. The analysis methods established for plants with relatively simple shapes under controlled conditions are not directly applicable to complicated images from the field. Large variations in the target plant, light, and background need to be covered in annotating the images from the field to prepare a training dataset for DNN, which can be laborious when performed manually for thousands of target plants[14]. In addition, the best DNN architecture depends on variations in the images. Various DNN architectures are available with different strengths. For example, U-Net has been used for the segmentation of plants from the top view[6,13,15] as well as roots in soil[16]. Other architectures that are successful in other disciplines can also be promising for plant image analysis, e.g., SINet in segmenting camouflaged animals[17] or DANet in detecting fine structures such as human veins[18]. Thus, to efficiently analyze complex images from a field within a realistic workload, it is necessary to reduce the manual annotation effort and select the DNN architecture whose strength best suits the analysis of the features in the target images[6,19]. The application of DNN to high-resolution image analysis of plants in the field while overcoming the challenges described in this and the previous paragraphs enables the identification of diverse biological questions, including ecology and evolution, with pigment accumulation in allopolyploids and their progenitors being an example.

Accumulation of anthocyanin pigment is induced by various environmental conditions and is considered a stress marker[20–22]. Laboratory studies of the model plant *Arabidopsis thaliana* have shown that anthocyanin in leaves increases in response to various external stresses, such as intense light, cold temperature, and drought, as a protection against oxidation, making the plant appear reddish[23,24]. In contrast to laboratory conditions, even in *A. thaliana*, little is known about the mechanism of pigment accumulation in complex field environments in which air temperature, radiation, and precipitation fluctuate[25,26]. Furthermore, widespread variation of anthocyanin content within and among species suggests its evolutionary significance in adaptation and speciation[20].

Allopolyploid speciation occurs through hybridization between different species with genome duplication. Its prevalence among natural and crop plant species has stimulated discussions and debates regarding the advantages and disadvantages of allopolyploid species[27–29]. Since the end of the 20th century, a major focus of the polyploid study has been on genome-wide mutations that are induced at the time of polyploidization termed "genome shock". However, recent reports have shown a lack of genome shock in *Arabidopsis* and grass polyploids[30,31], suggesting that it is not essential for polyploid adaptation. Instead of novel mutations, environmental responses of diploid progenitor species can be inherited and combined in allopolyploid species, which was originally discussed in plant evolutionary and systematics studies[29,32–34]. Soltis et al. (2016) have emphasized that the paucity of model polyploid species that integrate functional and ecological data is a major barrier to testing evolutionary and ecological hypotheses on polyploidy[29,34]. In the model genus *Arabidopsis*, the allotetraploid species *A. kamchatica* is emerging as a model polyploid species, which was derived from two diploid progenitors, *A. halleri* and *A. lyrata*[35]. In addition to natural *A. kamchatica* genotypes, synthetic *A. kamchatica* plants can be used to examine the effects of environmental responses inherited from progenitors[36]. The natural distribution range of *A. kamchatica* is wider than that of diploid progenitors, both in latitude and altitude[37,38]. Physiological and transcriptome experiments in regulated laboratory conditions showed that *A. kamchatica* inherited the gene expression pattern associated with the cold response from the diploid progenitor *A. lyrata* that was distributed in colder habitats than the other progenitor[39,40]. From the diploid progenitor *A. halleri*, *A. kamchatica* inherited the gene expression patterns responsible for zinc hyperaccumulation and tolerance[41]. Zinc concentration analysis of soils from natural habitats showed that *A. kamchatica* can tolerate moderately contaminated soil, suggesting that the allopolyploid inherited adaptive environmental tolerance of *A. halleri* in natural fields[42]. In contrast to relatively stable natural environments such as soil metal concentrations, time-series field observations are critical for capturing plant reactions to fluctuating meteorological conditions.

In this study, we present PlantServation, a method for image acquisition and analysis that consists of hardware and software, and applied it to studying environmental and genotypic effects on pigment amounts using four *Arabidopsis* species as a case study. Using the robust yet inexpensive image acquisition system with an RGB camera, we collected daily images of small individuals grown in the field in Switzerland and Japan for five months each for three years. We developed an efficient image analysis pipeline using DNN by registering the position of individual plants in time-series images by augmenting annotation data and comparing the performance of multiple DNN architectures. We estimated the time-series anthocyanin content using leaf color information from PlantServation with experimental validation. We addressed two biological questions on the effect of environments and genotypes on anthocyanin pigments, respectively: (1) How do air temperature, radiation, and precipitation affect the anthocyanin content in *Arabidopsis* species in complex field environments? (2) Does the synthetic polyploid *A. kamchatica* recapitulate the seasonal fluctuation of anthocyanin in natural polyploids, and how is it associated with those of the diploid progenitors?

## Results

### Image acquisition in the field

We established an inexpensive image acquisition system that endured in the field for five months during the growing season over three years. The hardware part of PlantServation was set up in common gardens in Switzerland and Japan using commercial polytunnel skeletons and weather-resistant RGB cameras (RICOH WG-40) positioned at 150 cm from the ground to collect the top-view images (Fig. 1a). Normal camera batteries do not last longer than a few weeks in our environment. For a stable power supply and to save the labor of exchanging batteries frequently, we replaced camera batteries with custom-made direct current (DC) couplers that could be connected to a common power source with an alternating current (AC) adaptor (Fig. 1a, b). A commercial uninterruptable power supply (UPS) provides emergency power in unexpected power breaks (Fig. 1a). The use of a flat cable enabled a connection between the DC coupler and cable while closing and sealing the battery lid (Fig. 1a, b, Supplementary Fig. 1). The support bars were fixed to each other and to the polytunnel skeletons using moving scaffolding clamps, whereas the DC cables were fixed to

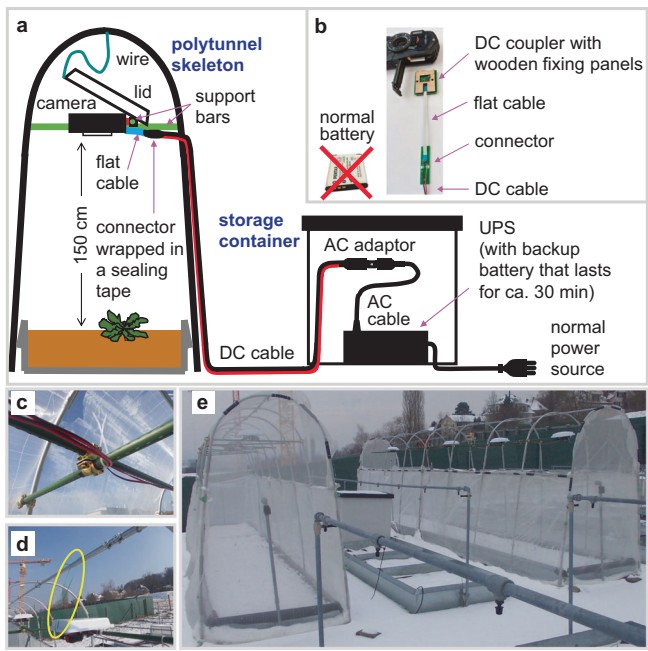

**Fig. 1 | The set-up for image acquisition. a** Illustration of the image acquisition system with a constant power supply. A wire connected to a lid was stretched out when the lid was lowered fully. Instead of a normal battery, a custom-made DC coupler was placed in a RICOH WG-40 camera using a flat cable. A UPS was installed as a temporal power source in case of a power cut. **b** Close-up of a DC coupler with wooden fixing panels (one for each side of the DC coupler), a flat cable, a connector, and a DC cable. A DC coupler was used instead of a normal battery. Wooden fixing panels and a DC coupler were inserted together in the camera battery slot to position the DC coupler stably. Details of the set-up are shown in Supplementary Fig. 1. **c** Support bars with DC cables fixed using cable ties and a movable scaffolding clamp with a black rubber sheet in between. **d** A wire (yellow oval) lifting the camera lid to avoid contact with the camera. **e** Overview of the image acquisition system. The polytunnel frames were covered with mesh, with the top mesh being removed during winter in Switzerland as seen in the photo.

the support bars using cable ties (Fig. 1c). To minimize the misalignment of cameras, we inserted a rubber sheet between the camera holder and support bar. We installed a custom-made lid to protect the cameras from snow, rain, and radiation (Fig. 1d). To avoid herbivores and birds, the polytunnel skeletons were covered with mesh sheets, with the top part kept open during winter to allow snowfall and to prevent damage to the polytunnel skeletons by seasonal winds (Fig. 1e). Altogether, the expense for the hardware part of PlantServation with eight cameras to observe 384 plants was ca. USD 2600 as of 2017.

Using PlantServation, we obtained approximately 4,032,000 images of target plants (12 genotypes × 20 replicates × 2 sites × 16–24 images/day × 150 days/year × 3 years) using the Interval Shooting function of the camera. These were captured using five cameras at each of the Swiss and Japanese sites. Each image had 16 M (4608 × 3456) pixels with a pixel range of 0 to 255 in the 8-bit sRGB color space and included the top view of the 48 target plants as one plot (Fig. 2a). The height and width of 1 pixel corresponded to approximately 0.45 mm. The 48 target plants consisted of four blocks, each of which consisted of 12 genotypes representing four species of *Arabidopsis* (Supplementary Table 1): the model species *A. thaliana*, natural and synthetic allotetraploid *A. kamchatica*, and its diploid progenitors *A. halleri* and *A. lyrata*[35,38,43–45]. Two independently synthesized *A. kamchatica* plants, along with their progenitor genotypes, enabled the comparison of polyploids shortly after emergence with those long after establishment and generations under natural selection, while the inclusion of the model plant *A. thaliana* facilitated the

interpretation of results in light of previous molecular and physiological studies[40].

The acquired images contained large variations in layout, background, and target plants. The layout of the frame around the plots varied between the Swiss and Japanese sites, whereas the background (soil, sand, and humus) varied between and within sites (Fig. 2a). Variation also existed at the target plant level with respect to light conditions, background, and the color, shape, and size of plants (Fig. 2b). In both sites, four white marble balls of 2.5 cm diameter, fixed to wooden or metal nail inserted in the ground, were placed at the corners of the rectangular area containing the target plants (Fig. 2a). These were used as markers for adjusting the position in the images to be consistent throughout the time points (see Step 2 in **Image analysis pipeline** for an overview and **Methods** for details). Using our economical image acquisition system, we successfully collected daily images of different genotypes of *Arabidopsis* in the field over several months.

## Image analysis pipeline

In the PlantServation software, images from each camera were processed using a pipeline implemented in Python (Fig. 3a). Here, we provide a brief overview of our pipeline and describe the details in the **Methods** section. In Step 1 (Fig. 3a), we selected up to four images per day by thresholding the pixel values and setting a time window close to midday to reduce the variation in brightness among the images, yielding ca. a total of 740,432 individual plant images.

In the field, the cameras inevitably move owing to wind and during maintenance, resulting in inconsistencies in the position of the target plants among the images. To address this issue, in Step 2, we performed registration and compilation of the images from different time points and defined the plant position as the peak where the center of the plant was most frequently detected (Supplementary Fig. 2).

Once the center of the plant was defined, we segmented the plants in Step 3 after cropping individual plants from the image using end-to-end segmentation with DANet which was the best among the five DNN architectures examined (Supplementary Fig. 3, Supplementary Table 2). We used a custom-made training dataset consisting of 7,500 images augmented from 225 manually labeled images (Supplementary Fig. 4).

We examined the performance of our pipeline using DANet by analyzing 30 plant images that were not used to build the pipeline and by comparing the outcome with the ground truth, the plant area marked by humans. The pipeline worked reasonably well in segmenting different background types at the Swiss (soil or sand) and Japanese (sand and humus) sites (Supplementary Table 3). The performance of the segmentation was slightly higher for the soil background (Fig. 3b). When the color of the plant and that of the background were similar, the Dice coefficient was lower; however, this did not hinder the subsequent color analysis, the major purpose of this study (Fig. 3b).

In Step 4, we color-converted the pixel values in RGB of the segmented plant area to those in L*a*b* (Fig. 3c). We subsequently calculated the average value of L*a*b* per target plant. Supplementary Movies 1–3 show examples of time-series compilation of a segmented plant area.

Finally, the pipeline outputs color information, genotype, and date for each target plant. We excluded anomalous data from further analyses by referring to the field record, for example, on snow cover and plant death, by a *z*-score threshold calculated with nearest neighbor interpolation and by visually examining the original image for individuals whose time-series plots deviated from the norm (Supplementary Fig. 5; see **Methods** for further details).

## Validation of color-based estimation of leaf pigment content

To estimate the anthocyanin content from the color information of images, we used another set of plants to collect color information from

**a**

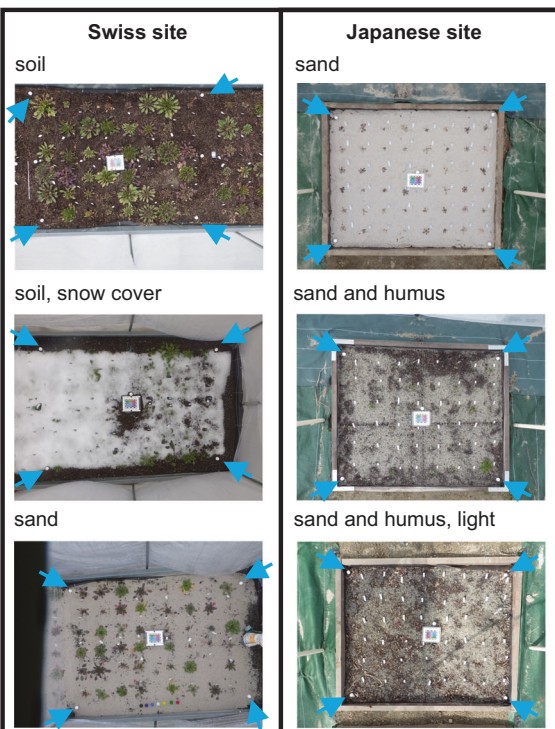

**b**

**Fig. 2 | Examples of images of *Arabidopsis* collected in the field. a** Original images from the Swiss site (left column), and from the Japanese site (right column). Background (soil, sand, humus), surroundings (mesh around the plot, frames around the plot, or sheet on the ground), and the amount of noise (e.g., snow seen in the middle images from the Swiss site, strong light on the right half of the bottom images from the Japanese site) varied among images. Blue arrows indicate white marbles placed at the corners of the target area. **b** Close-up images representing the diversity in light conditions, in the background, and in the color, shape, and size of plants.

images and compared it with the actual pigment content measured experimentally. We obtained both anthocyanin content per leaf weight and per leaf area and compared models with different color spaces, indices, and model types (see **Methods** and Supplementary Figs. 6 and 7). Among model types, a random forest model, a machine learning method, performed the best, consistent with a previous study of pigment estimation from color in *Arabidopsis* in the laboratory[46] (see Supplementary Fig. 6 and **Methods** for details). Among the random forest models, the model with the relative anthocyanin content per leaf weight as a response variable and L*, a*, and b* as explanatory variables performed the best (Supplementary Fig. 7). Among the three color features, a* and b* largely contributed to the variation in anthocyanin content per weight, followed by L* (importance: a* 40.5%, b* 43.3%, L* 14.5%). Fitting results indicated that there was a high correlation between measured and estimated anthocyanin content per weight when genotypes were separated (Pearson correlation coefficient $r > 0.850$ for the majority of genotypes, Supplementary Fig. 8) and pooled (Pearson correlation coefficient, $r = 0.846$, $p < 2.2e-16$, Fig. 4, Source Data 1).

For anthocyanin content per area, we found that the same method worked the best, although its R-squared value ($R^2$) was lower than that of anthocyanin content per weight (Supplementary Fig. 7). The fitting result showed a similarly high correlation between the measured and estimated values, although the values are overestimated in the small value range and underestimated in the large value range (Supplementary Figs. 9 and 10, Source Data 1). In the subsequent analyses on the effect of environments and genotypes, both anthocyanin content per weight and per area resulted in similar results, and we will show the former in the main text.

We also measured the chlorophyll content of the same leaves, which also affects leaf color[47]. Anthocyanin per weight was negatively correlated with chlorophyll per weight, however, the variation was large, suggesting that no clear dependency of the anthocyanin content on the chlorophyll content (Pearson correlation coefficient, $r = -0.433$, $p < 2.2e-16$, Supplementary Fig. 11). Although it is possible that the chlorophyll content works as noise in estimating anthocyanin content, these analyses suggest that the leaf images contained information of the anthocyanin content. Thus, we next examine whether our estimation model works in capturing the seasonal patterns of anthocyanin fluctuation and in evaluating the effects of environments and genotypes.

**Seasonal fluctuation of plant traits**

We applied a random forest regression model in the previous section to the time-series images and estimated the anthocyanin content from L*a*b* in the image per time point per genotype per site. To examine the pattern of fluctuation in the estimated anthocyanin content and other traits in the field and the variation among genotypes, we generated time-series plots for each trait per site per genotype (Fig. 5 and Supplementary Figs. 12–15, Source Data 2–7). The values were averaged among images when there were multiple images per day.

The variation in the estimated anthocyanin content throughout the season and among genotypes was more pronounced at the Swiss site than at the Japanese site (Fig. 5 and Supplementary Fig. 12). At the Swiss site, the anthocyanin content of most genotypes increased from autumn to winter. The difference between the two sites is consistent with the mild winter conditions at the Japanese site, in which snow rarely falls. These data will be used for analyses in the following two sections.

The a* trend resembled that of the estimated anthocyanin content (Fig. 5 and Supplementary Figs. 12 and 13). In contrast to a* and the estimated anthocyanin content, b* and L* showed a reverse pattern

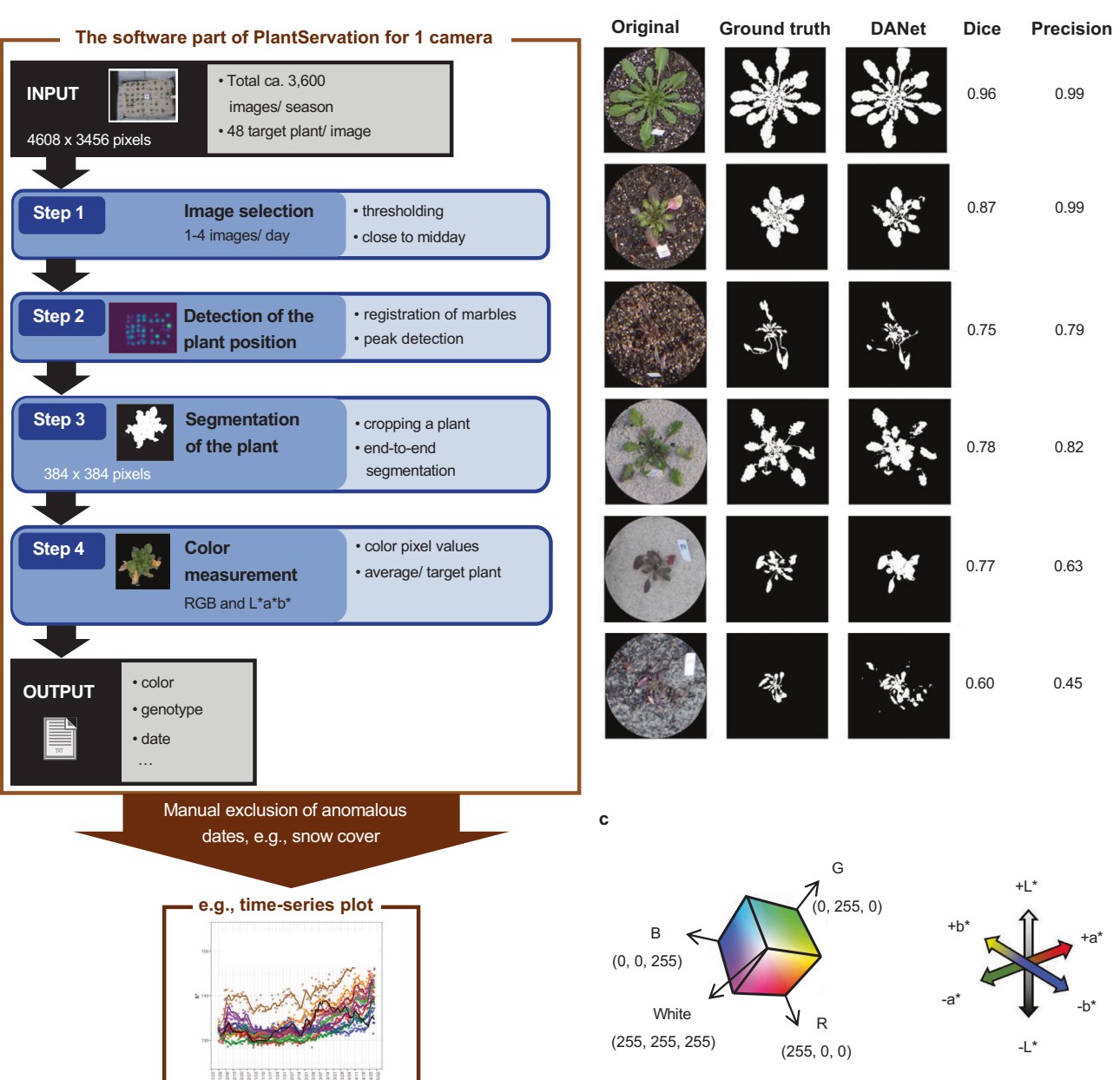

**Fig. 3 | Image analysis pipeline and its performance. a** The workflow of the image analysis. 16–24 images per camera were taken per day for ca. 150 days per season. Images from five cameras per site for three seasons were subject to the analysis. **b** The evaluation result of the outcome of the machine learning using DANet. Images varying in plant size, shape, color, and background were examined concerning the Dice coefficient and Precision. Six representative images are shown. For each plant, the original image with circular mask, ground truth, DANet segmentation outcome, and the score for Dice coefficient and Precision are shown. From top to bottom: a green plant with a soil background in the Swiss site, a plant with green and dark leaves with a soil background in the Swiss site, a dark plant with a soil background in the Swiss site, a green plant with a sand background in the Swiss site, a dark plant with a sand background in the Swiss site, a dark plant with a humus background in the Japanese site. **c** RGB color space (left) and L\*a\*b\* color space (right).

over time and in the order of the genotypes according to the value of the variable (Fig. 5 and Supplementary Figs. 12–15). These results were consistent with that a\* ranges along the green–red gradient and thus corresponds to the leaf color gradient well (see the previous section and Fig. 3c).

**Effect of environmental factors on anthocyanin content**
Experiments in regulated chamber conditions showed that anthocyanin in *A. thaliana* is induced by stress treatment at low temperatures, strong light, and drought[23,24]. Using the time-series anthocyanin content estimated in the field, we examined whether environmental conditions had a significant effect on the anthocyanin content of *A. thaliana* and its relatives in complex natural environments (Fig. 6a, b). We fitted linear regression models with the estimated anthocyanin content as the response variable and radiation, coldness, and precipitation as explanatory variables. Considering the response time and threshold of plants to environmental cues, we adopted the best parameter combination for window, lag, and temperature thresholds in the past month, as in previous phenological studies[48] (Fig. 6a). We found that the estimated anthocyanin content was associated strongly

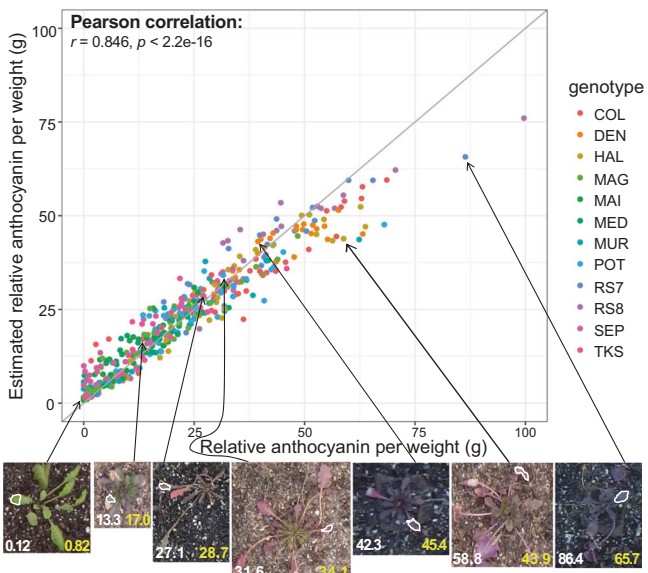

**Fig. 4 | Scatter plot of relative anthocyanin content per weight and estimated relative anthocyanin content per weight in leaves of 12 *Arabidopsis* genotypes using random forest model.** The gray line indicates $y = x$. Images below the plot show representative examples of the estimation of the anthocyanin content using a random forest model. White and yellow numbers indicate measured and estimated values, respectively. The part of a leaf subject to the analysis is surrounded by a white line. In the time-series image analysis, the whole plant area was analyzed using the random forest model built here. Pearson's two-sided correlation test: $r = 0.846$, $p < 2.2e-16$. $n = 451$. Source data are provided as a Source Data file.

with coldness and radiation and relatively weakly with precipitation in most genotypes (Supplementary Tables 4 and 5). This suggests that the three stress conditions studied in the laboratories also exhibited significant effects in the field conditions and confirmed that Plant-Servation workflow is useful for evaluating the environmental effect on plant traits. In addition, the analysis showed that each of the three environmental factors explained up to 60% of the variation in the estimated anthocyanin content. At the Swiss site, the radiation was particularly influential (Fig. 6b and Supplementary Fig. 16, Source Data 8). Time-series plots of the model species *A. thaliana* at the Swiss site with the estimated environmental parameters suggested that the estimated anthocyanin content was well associated with radiation, whereas the relationship between rainfall and coldness was not straightforward (Supplementary Figs. 17 and 18, Source Data 8). At the Japanese site, coldness contributed significantly to all the genotypes, although the number of significant factors tended to be lower than those at the Swiss site, possibly because of the mild winter with negligible snow. Considered together, the results suggest that coldness, radiation, and precipitation contribute to the anthocyanin content in the field, with the extent of contribution varying among environmental factors, sites, and genotypes. Roughly, half of the variations in anthocyanin content were not explained by the three environmental factors, suggesting the need for further studies on other factors and their combinatorial effects (see **Discussion**).

### Anthocyanin content in *Arabidopsis* polyploids
Next, we examined the differences in the estimated anthocyanin content between species and genotypes. We addressed the question of whether synthetic polyploids recapitulate the patterns of natural polyploids and whether they combine the traits of diploid progenitor species. We included two independent synthetic polyploid genotypes of *A. kamchatica* with their diploid progenitors: RS7 derived from HAL (*A. halleri*) and SEP (*A. lyrata*), and RS8 derived from HAL and MED (*A. lyrata*). As natural *A. kamchatica*, we planted four genotypes of

Japanese polyploids and two genotypes named Northern polyploids, which are estimated to have originated independently from Japanese polyploids[38]. The Swiss site showed a higher variation among genotypes and seasons, as described above, and the genotypes can be grouped roughly into four according to the pattern of fluctuation of the estimated anthocyanin (Fig. 5 and Supplementary Fig. 12). First, the diploid progenitors *A. lyrata* (MED and SEP) had a higher content than others in early seasons, followed by intermediate in late seasons. Second, the diploid progenitor *A. halleri* (HAL) showed an opposite trend (intermediate and subsequently higher than the others). Third, the Japanese allopolyploids exhibit low contents throughout the seasons. Fourth, synthetic allopolyploids and natural allopolyploids of northern origin showed an interesting pattern. They did not constitute a simple mean of diploid progenitors but a combination of periods during which they resembled diploid progenitors (Fig. 5 and Supplementary Fig. 12). For example, the synthetic allopolyploids showed a trend similar to that of the natural diploid *A. halleri* at the beginning and to the diploid *A. lyrata* later in the second year at the Swiss site. Overall, the trend of the synthetic allopolyploids resembled that of the diploids with a smaller content of the estimated anthocyanin at a given time point.

To compare the trend of the estimated anthocyanin content among genotypes throughout the seasons, we conducted dimension reduction via principal component analysis (PCA) on the estimated anthocyanin for all years from both sites (Fig. 7 and Supplementary Fig. 19, Source Data 9). The first principal component (PC1) explained 51.8% of the variation in the data (Fig. 7). Along PC1, the diploid progenitors were closer to the synthetic allopolyploids, and the synthetic allopolyploids were closer to the natural allopolyploids of northern origin than to the natural allopolyploids from Japan (Fig. 7). The second principal component (PC2) explained 19.7% of the variation. Along PC2, two diploid species were located at both tips (Fig. 7).

The PCA plot indicated that the two independent synthetic polyploids showed a similar pattern, confirming reproducible changes immediately after polyploidization. Furthermore, the synthetic polyploids were closely related to natural polyploids, that is, two genotypes of northern polyploids (Fig. 7 and Supplementary Fig. 19). The other four natural polyploids showed similar PC2 values to synthetic polyploids but diverged at PC1, potentially reflecting their different or older polyploid origin (see **Discussion**). These data suggest that the synthetic polyploids recapitulated the pattern of anthocyanin responses of certain natural polyploids. In addition, the synthetic polyploids resembled one of the parents *A. lyrata* in PC1 and another parent *A. halleri* in PC2 (Fig. 7). When the data of the two sites were analyzed separately, a similar pattern was found for the Swiss data, and the synthetic polyploids were located in the middle of the two diploid progenitor species both in PC1 and PC2 for the Japanese data (Supplementary Figs. 20 and 21, Source Data 11). Moreover, the PCA plot of a* for all years from both sites resembled those of anthocyanin content (Fig. 7 and Supplementary Figs. 19 and 22, Source Data 9). These analyses support the observations of the inheritance and merger of the traits of the two diploid progenitors. The grouping of the genotypes in these analyses on genotype averages was generally consistent with that of the analyses at the individual plant level (Supplementary Figs. 23 and 24, Source Data 12). The plots of the PC scores with the estimated anthocyanin content for each genotype suggest that some site-by-year combinations, such as the Swiss site Year 1 (yr1) for PC1 and the Swiss and Japanese sites Year 3 (yr3) for PC2, may be particularly influential to the among-genotype difference of the overall PCA plots (Supplementary Figs. 25 and 26).

## Discussion
Using PlantServation, we successfully distinguished the time-series trends of different plant genotypes and species varying in morphology, color, and size under fluctuating and noisy outdoor conditions via

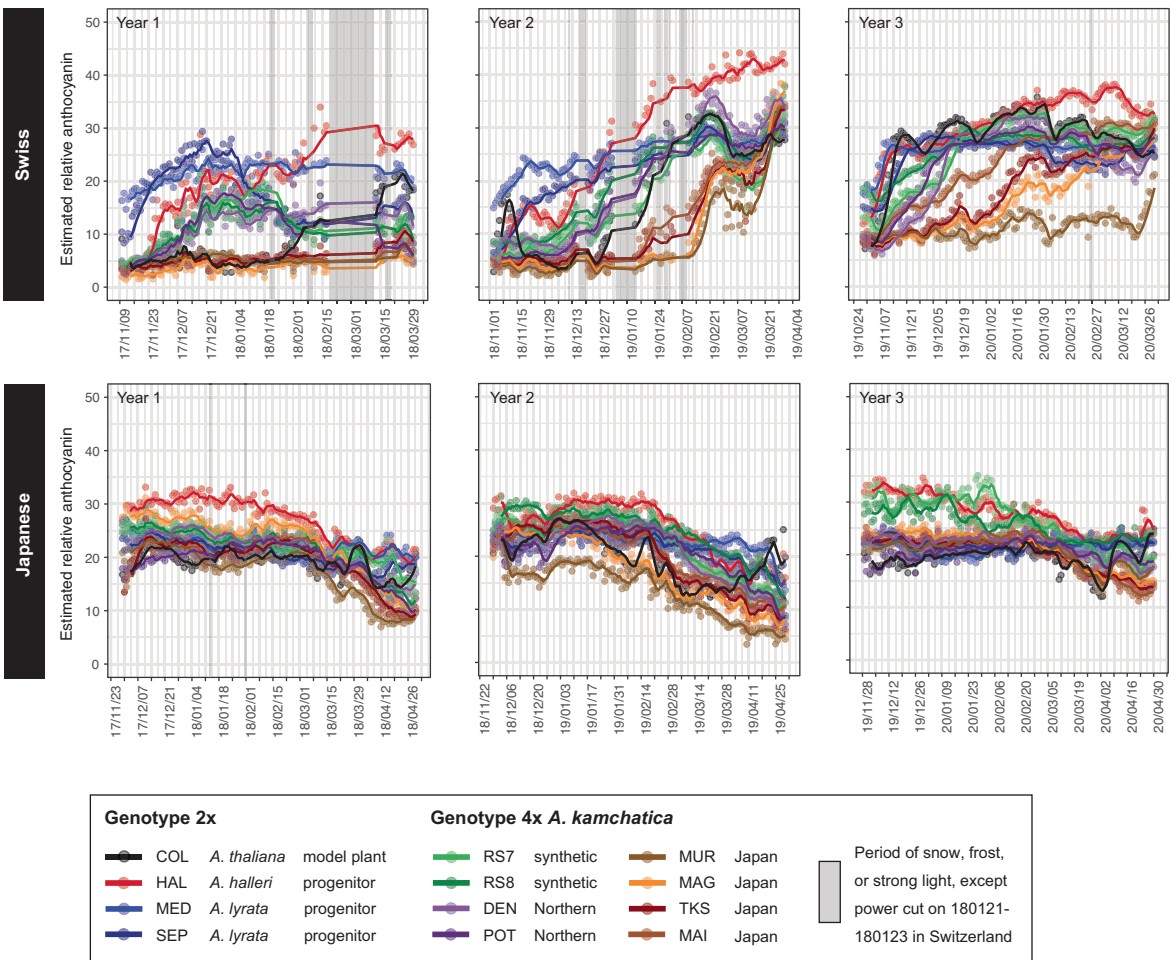

**Fig. 5 | Time-series plots of the 5-day moving average of estimated relative anthocyanin content per weight in plants of 12 *Arabidopsis* genotypes at the Swiss and Japanese sites in three seasons.** The data points indicate the arithmetic mean. The number of plants is summarized in Supplementary Table 8. Source data are provided as a Source Data file.

image analysis using DNN. This was achieved by a relatively small manual annotation effort owing to augmentation (Supplementary Fig. 4). Our data demonstrate that it is possible to quantitatively and non-destructively evaluate plant color at high-resolution throughout the season. Hence, we could overcome the challenges in conventional manual data collection and qualitative color evaluation using visual inspection or color charts[49]. Notably, our image acquisition system is robust and simultaneously relatively simple and inexpensive, allowing non-experts to implement the data acquisition without a large investment. Many waterproof, shockproof, freezeproof, and dustproof cameras with interval shooting functionality are available on the market. Any of such tough cameras can be an alternative if a compatible DC coupler can be prepared. All other parts, e.g., support bars, cables, terminals, and UPS, of the hardware are also easily available at affordable prices. In this regard, this study paves the way for comparing and classifying the responses of plants of different genotypes and species to seasonally fluctuating environments.

The best DNN architecture for analyzing our dataset was DANet. This could be attributed to the strength of DANet in detecting fine structures such as petioles in our dataset. For our dataset, DANet outperformed SINet, though the latter is widely known for detecting camouflaged objects. Although not optimal for our dataset, U-Net has been shown to perform effectively with small labeling datasets in medical research[50–52] and has recently been increasingly applied to agriculture for leaf disease symptom diagnostics[53]. Many DNN architectures have been developed in fields other than plant science[11]. When

selecting a DNN architecture, widening the search beyond plant science can yield a better solution. Indeed, DANet was originally developed to detect scenes and objects on the street in computer vision and was later successfully applied to detect fine vessels in the human retina in medical research[18,54]. For accurate segmentation, it is important to grasp the critical features of the target image and select a suitable DNN architecture that can detect them, regardless of the type of object to be segmented.

Augmentation achieved reasonable segmentation performance with a relatively small labeling dataset. We used 225 manually labeled images, which is markedly less compared with similar indoor studies in which hundreds or thousands of images were labeled[12,13]. Considering the complexity of our images from the field, our approach was highly labor-saving. For similar plant segmentation tasks in future studies, fewer manually labeled data will be sufficient to construct a DNN model using the transfer learning of our learned model as well as the data augmentation strategy[55]. Our data also highlight many difficulties in handling time-series images from the field. Among them, the detection of the target plant was the most critical for accurate data acquisition for downstream analysis. In particular, whenever the color and texture of the target plant and background are similar, it is challenging to identify the area of the target plant, even for humans. Even though it might be difficult to avoid such situations, data quality could be improved by addressing other issues. A potential future possibility is to change the segmentation procedure. Segmentation was independently performed for each frame. The incorporation of temporal

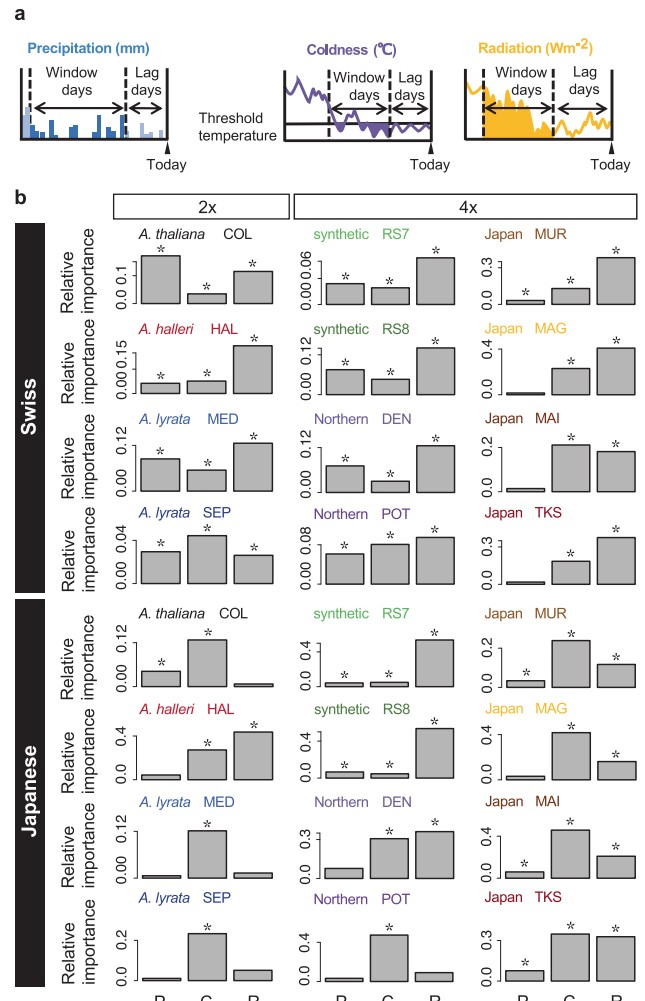

**Fig. 6 | Environmental variables associated with the estimated anthocyanin content per weight. a** Illustration of the concept of the moving total of precipitation, coldness, and radiation used in the regression analysis. **b** Relative importance of the environmental variables on the estimated anthocyanin content for 12 genotypes of *Arabidopsis* at the Swiss and Japanese sites. The genotype code in three capital letters is indicated for each genotype. 2x indicates diploid and 4x indicates allotetraploid *A. kamchatica*. P: Precipitation, C: Coldness, R: Radiation. Significant variables based on confidence intervals calculated with bias-corrected and accelerated (BCa) bootstrapping are indicated with asterisks. $n = 663$ for the regression analysis of each genotype. Source data are provided as a Source Data file.

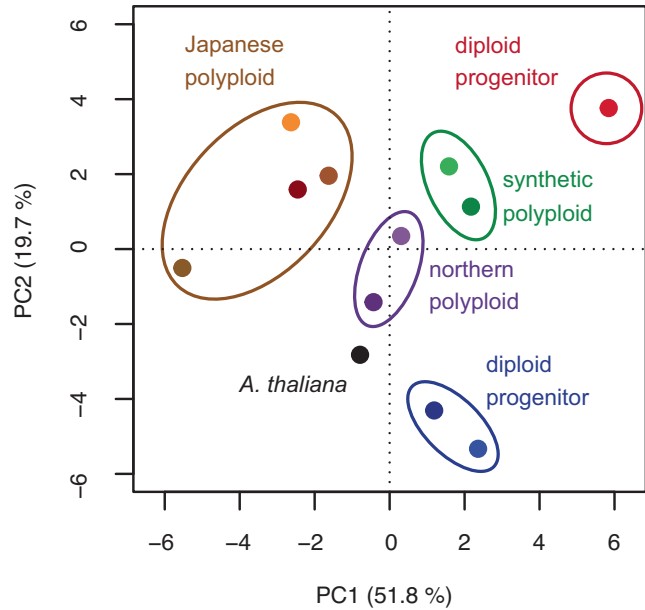

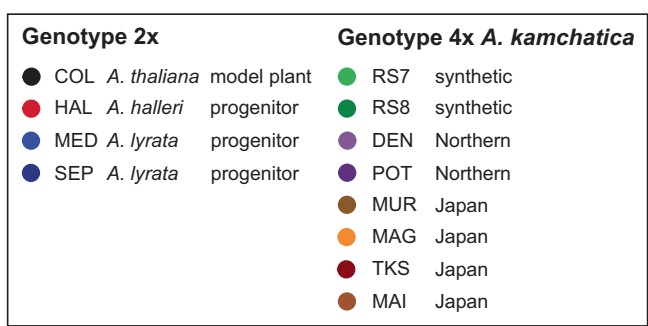

**Fig. 7 | Principal component analysis plot of time-series estimated relative anthocyanin content per weight for 12 *Arabidopsis* genotypes.** The data from the Swiss and Japanese sites in three seasons are merged. Source data are provided as a Source Data file.

information into the DNN input may facilitate the segmentation of images that are difficult to segment within a single frame. In this context, the application of Video Object Segmentation methods is potentially promising[55]. Another possibility is the additional implementation of manually obtained information regarding the target plant and its circumstances. Non-target objects in the image, such as snow, can be labeled for training the DNN to distinguish them from the target plant during segmentation (Supplementary Fig. 5a). Cross-referencing with manual records is effective for issues that are difficult to address by modifying the analysis pipeline. For example, plant death may not be precisely recognized by the DNN when the plant body remains intact (Supplementary Fig. 5b). In such cases, manual scoring of plant survival can complement the analysis of time-series growth.

Overall, time-series image acquisition in the field inevitably faces a number of predictable and unpredictable challenges, from the similarity in color and texture between target plants and their surroundings to the interference of non-target objects. Not all the issues would

be solvable; however, a priori preventive measures as well as the measures necessary for the image acquisition system should improve the quality of downstream image analysis. An effective understanding of the study system, e.g., the plant life cycle and weather conditions at the study site, would enable the selection of the appropriate measures for each case.

We estimated the time series of leaf anthocyanin content from the color information. As in a previous study on *Arabidopsis thaliana* in the laboratory, a random forest model functioned effectively for our dataset in the estimation of anthocyanin[46]. With the strength of capturing nonlinearity in data, a random forest model can be effective for samples from the field where the environment is heterogeneous, causing noise in the color information. By successfully estimating anthocyanin content from color information, this study opens up the possibility of monitoring plant physiological responses to the environment over months in a non-destructive manner.

Our study demonstrates that time-series data can contribute to the evaluation of plant responses to environments. Regression analyses indicated that in most genotypes, more than one of the cumulative coldness, radiation, and precipitation in the past days to weeks were significantly associated with the estimated anthocyanin content in the field. These results are consistent with those of previous studies in the laboratory reporting temperature, light, and drought as key environmental factors affecting anthocyanin accumulation and highlight the importance of considering multiple environmental factors

(Fig. 6b)[2,23,24,56]. The linear regression model incorporating lag and window of past temperature suggests that the anthocyanin content is better explained by past temperatures than by the current temperature (Supplementary Tables 4 and 5, the extreme would be lag 0 and window 1). Along with gradual changes in anthocyanin content, plants may accumulate past temperature information through transcriptional and epigenetic memory similar to vernalization, seasonal responses, or heat acclimation[1,57–60]. The observed diversity in anthocyanin fluctuations may have been caused by the diversity in the sensitivity of such environmental responses as well as the regulatory variation of pigment metabolic enzymes. Integration of transcriptome and metabolome data with image analysis is valuable for understanding the molecular basis of plant responses *in natura*.

The three environmental factors that were tested based on previous laboratory studies explained roughly half of the variation in the anthocyanin content in the field. Besides noise in the field, the unexplained variations suggest that the linear regression models incorporating the lag and window of the three factors are not adequate for understanding the complex fluctuations in the field. Fine-grained measurement of climatic variables, such as soil moisture, distinct measurement of air and soil temperature, the spectrum of irradiation including UV-irradiation, and the interaction thereof, would explain more variations in anthocyanin content[2,21]. In addition, biotic interactions such as disease and herbivory are known to affect anthocyanin content, although they were not observed in these experiments[21,22]. We suggest that little is still known about plant environmental responses *in natura*. Our pipeline paves the way to search for combinatorial environmental effects along with intrinsic developmental stages and thus, would increase the proportion of anthocyanin changes to be explained.

Time-series monitoring of anthocyanin content and leaf color in *Arabidopsis* species in the field provided a unique opportunity to address long-standing questions on polyploidy using ecological data[29,34]. First, the fluctuations in anthocyanin content and leaf color of the synthetic allopolyploid *A. kamchatica* in the field were highly similar to those of certain natural polyploids, that is, northern polyploids. The similarity of synthetic allopolyploids to a subset of natural counterparts that exhibit variation in a trait is consistent with the findings on the pigments (cyanidin, quercetin, and kaempferol) and color in *Nicotiana* flowers under controlled conditions[61,62]. Our time-series data suggest that synthetic polyploids can recapitulate the polyploid speciation that is observable in the fluctuation of anthocyanin content and leaf color in outdoor conditions. The wide variation among the natural polyploids may be attributed to their local adaptation or independent origins. Molecular population genetic studies suggested that northern polyploids of *A. kamchatica* had an independent polyploid origin and possibly originated more recently than Japanese polyploids based on their high similarity to diploid sequences[38,63]. The divergence of Japanese polyploids from synthetic polyploids may be attributed to the longer evolutionary time since polyploidization. In addition, Japanese polyploids may originate from diploid genotypes with different anthocyanin responses.

Second, the analysis supported a long-standing hypothesis stating that the synthetic polyploid can combine the responses of two progenitor species. The two independent synthetic polyploid lines showed similar results, suggesting that the stochastic novel mutation at the polyploidization events (may be called "genome shock") did not play a significant role[29]. This phenotypic trait analysis is in agreement with transcriptomic studies showing inherited and combined responses of allopolyploid species[29,40,64,65]. Interestingly, the anthocyanin content of the synthetic polyploids tended to be closer to the lower values of the two diploid progenitors. If we can assume that anthocyanin content is a stress indicator, it suggests that synthetic and northern *A. kamchatica* accumulate less anthocyanin than progenitors

because they are able to withstand stress throughout the season by obtaining a generalist niche through the combination of progenitors' environmental responses. Consistent with the observed similar anthocyanin contents of *A. kamchatica* and *A. lyrata* during cold seasons, laboratory studies suggested that *A. kamchatica* inherited cold tolerance and associated gene expression patterns from the progenitor *A. lyrata*[39,40]. To further verify this generalist hypothesis originally proposed by Stebbins, measurements of fitness components are important[32].

PlantServation enables continuous image acquisition in the field over months, capturing changes in leaf color and pigment in seasonally fluctuating environments, while retaining the simplicity of the image acquisition system reported earlier[66]. As laboratory settings do not necessarily reproduce field environments, field data are essential for understanding plant responses to complex natural environments[3]. We envisage the application of PlantServation in field experiments on model species. The spatial resolution of PlantServation enabled the monitoring of *Arabidopsis* seedlings at the individual level. Field observations of *A. thaliana* and its relatives will enable the study of plant responses in the field, taking advantage of a large number of studies in regulated chambers as well as mutant collections. For example, destructive sampling of *A. thaliana* showed that anthocyanin content was reduced in the double mutants of *UVR8* and *CRY1* photoreceptor genes in the field[67], which can be extended to time-course studies. By connecting to solar power, PlantServation can be deployed at remote natural sites to observe diverse species. Furthermore, PlantServation can be exploited for other studies, including the screening of crops, where trait scoring at an individual plant or a finer level is informative. For example, the drought and iron deficiency responses or disease resistance quantified in previous studies can be monitored for a longer period[68–70]. Furthermore, although it was not the focus of the current study, the combination of our image acquisition system and the image analysis pipeline could also detect differences in morphological features among plants, as observed in the segmentation results (Fig. 3b). Thus, this study should also contribute to fine-scale morphological analyses using field images in future studies.

In conclusion, we demonstrated that it is possible to estimate the anthocyanin content in plants grown in the field based on the color information obtained from images using an inexpensive photoshooting system and an efficient image analysis pipeline using DNN. Moreover, we showed that time-series field data can provide insights into plant evolution and environmental responses, furthering our understanding of how plants thrive *in natura*.

## Methods
### Study species
We studied 12 genotypes of four species in the genus *Arabidopsis*. The *Arabidopsis thaliana* ($2n = 10$) Col-0 accession was used as the standard experimental strain. *Arabidopsis lyrata* ($2n = 2x = 16$) inhabits a circumpolar region, whereas *A. halleri* ($2n = 2x = 16$) is distributed in central Europe and the Far East[43]. The allotetraploid *A. kamchatica* ($2n = 4x = 32$) originated from diploids *A. lyrata* subsp. *petraea* and *A. halleri* subsp. *gemmifera*[38]. *Arabidopsis kamchatica* is widely distributed ranging from Taiwan, Japan, and Siberia in the Far East to North America[38]. Two subspecies are recognized in Japan: *A. kamchatica* subsp. *kawasakiana*, which is restricted to sandy shores in the lowlands and *A. kamchatica* subsp. *kamchatica*, which occurs at various altitudes[35,38,44]. Two synthetic *A. kamchatica* samples of laboratory origin were included (see Supplementary Table 1 for details). The genotype RS8 was synthesized by applying colchicine to a seedling of a hybrid of the diploid progenitors *A. halleri* subsp. *gemmifera* (W302 from Japan, called HAL) and *A. lyrata* subsp. *petraea* (named MED after its synonym *Arabis media*); the genome assembly has been reported for both[63,71]. The genotype RS7 was spontaneously polyploidized from

a hybrid of the same *A. halleri* genotype and *A. lyrata* subsp. *petraea* from another population in Siberia (named SEP after its synonym *Arabis septentrionalis*)[72]. The number of generations since allopolyploidization was three to five for a given year (yr) of the experiment, with four (yr1 and yr2) and five (yr3) for RS7 and three (yr1), four (yr2), and five (yr3) for RS8, respectively. All the genotypes were propagated by selfing in a laboratory.

## Study sites
The study was conducted at the Center for Ecological Research, Kyoto University, Japan (34°58′16″N, 135°57′24″E, 150 m a.s.l.), and at the University of Zurich, Switzerland (47°23′46″N, 8°33′05″E, 508 m a.s.l.).

## Experimental set-up
The experiment was conducted from autumn (seedling stage) to the following spring (end of the vegetative stage), starting in 2017 (yr1), 2018 (yr2), and 2019 (yr3). According to the local phenology of the study species, we transplanted seedlings in Switzerland and Japan in September or October, and November, respectively.

## Plant cultivation
**Swiss site**. Seeds of *Arabidopsis lyrata*, natural and synthetic *A. kamchatica*, and *A. thaliana* were sown in 12 well plates (Nunclon ™ Delta Surface, Thermo Scientific, Denmark) with quartz sand (0.1–0.8 mm TOP MINERAL AG, Switzerland) hydrated with tap water and placed at 4 °C for a week and subsequently by a window to stimulate germination. The seedlings at the cotyledon stage were transferred to biodegradable pots (W × L × H: 3 cm × 3 cm × 5 cm Peat Pot Strips, Jiffy) filled with a mixture of Floratorf (Floragard, Oldenburg, Germany): quartz sand (0.4–0.8 mm Quarzsand, Carlo Bernasconi AG, Switzerland) = 1:1 in volume. The potted seedlings of *A. lyrata* and *A. kamchatica* were cultivated in a growth chamber with a long-day setting (22 °C/20 °C, 16 h:8 h light: dark, RH 60%, light 120–140 µE) for six weeks, while *A. thaliana* seedlings were placed in a short day chamber for three weeks to keep them vegetative (18 °C/16 °C, 8 h:16 h light: dark, RH 60%, light 120–140 µE). For *A. halleri*, we cultivated clonally propagated small branch segments (1–2 cm) in pots in a long-day chamber for five weeks. For all species, potted plants were placed on a plastic tray covered with a transparent lid that was half-opened on the 3rd day and fully opened one week after potting. At potting and once per week, watering was performed using Wuxal Universaldünger nutrient solution (Maag, Westland Schweiz GmbH, Switzerland). We acclimated all plants outside under the roof for a week before transplanting them to the common garden at the Irchel Campus of the University of Zurich. The plantlets were planted in two built-in compartments, each 1 × 7 m, filled with well-watered Rasenerde (Ökohum GmbH, Switzerland). Only in yr3 we had a ca. 5 mm layer of quartz sand on the soil surface to ensure better color contrast between plants and the background. The entire compartment was covered with polyester mesh sheets (1 mm × 1 mm grid) over the skeleton of the polytunnel (model A-17, Nan-ei Kogyo, Japan) to prevent herbivory and damage by birds. During winter, the mesh at the top was removed to allow snowfall and prevent the skeletons from collapsing owing to strong winds.

**Japanese site**. The seeds of *A. lyrata*, natural and synthetic *A. kamchatica*, and *A. thaliana* were sown on sterilized plastic dishes (Asnol Petri Dish φ90 × 20 mm, 1-8549-04, AS ONE Corporation) filled with ca. 60 g of 0.3-0.6 mm quartz (#5, Toyo Matelan Co. Ltd.) hydrated with tap water. The dishes were kept in an incubator (KOITOTRON HNM-S11, KI Holdings Group) at 21 °C/15 °C, 12 h:12 h light: dark for four weeks, except for *A. thaliana* which was kept there for one week. When the seeds did not germinate after four weeks, the dishes were kept in the chamber for up to additional two weeks. The seedlings at the cotyledon stage were planted on blocks of mineral wool (rock fiber for cultivation M40T40, Nittobo) that were kept in an incubator (22 °C/

20 °C, 16 h:8 h light: dark, RH 74%, light 125–145 µE, KOITOTRON HNM-S11) for six weeks, except the branch segments of *A. halleri* prepared in the same manner as in the Swiss site and the seedlings of *A. thaliana* which were brought in one week later and three weeks later, respectively. Watering was performed twice a week, once of which using × 2000 HYPONeX solution (HYPONeX Japan Corp., Ltd.). All seedlings were acclimated for a week outside under the roof before transplanting. The seedlings were transplanted in a compartment (W × L × H: 100 cm × 100 cm × 20 cm) filled with a mixture of 40 L of humus (100% Shinshu Fall Leaves 100% Natural Fermented Products, Koshin Kawara Co. Ltd.) and 60 kg of the 0.3-0.6 mm quartz. We covered the surface of the soil mixture with a 1–2 mm thick layer of 0.3–0.6 mm quartz to increase the color contrast between the plants and the ground. The ground was wet immediately before transplanting and watered once directly after transplanting to promote the plant establishment. During the growing season, we set up a cage with mesh (20 µm × 20 µm, Moritaya, Japan & Marushin, Japan) over each compartment to prevent herbivory. A root-cutting-sheet ('Kurapapy', Kuraray Co. Ltd. in yr1 and yr2 and 'Paopao Nekiri Sheet' Nihon Nougyou System in yr3) was placed at the bottom of the soil to prevent overgrowth.

## Experimental design
The plants were placed according to a randomized complete block design with a 15 cm interval to the next plant. Twelve plants consisting of twelve genotypes were randomly assigned within block. Four adjacent blocks (2 × 2) constituted one plot, each of which was monitored using a single camera. The number of plots was five for each of the Japanese and the Swiss sites.

## Image acquisition (PlantServation hardware)
We used a RICOH WG-40 camera resistant to water, shock, dust, and freeze (protection level IP68) with autofocus and no flash modes. The detailed settings of the photo shoot are shown in Supplementary Table 6. The cameras were fixed onto frame bars using mounting tools (RICOH O-CM1472 and O-CH1470) that were specific to the camera. The acquired images were stored on an SD card and manually downloaded to a PC using a provided cable connection.

Continuous interval shooting throughout the season (16 images per day in yr1 and yr2 with 90-min intervals, and 24 images per day in yr3 with 60-min interval) was enabled by the customized power supply system (Fig. 1a). The combined use of an AC adaptor (9 V) and a house-designed DC coupler converted the voltage to 5 V to operate the camera (Fig. 1b, see Supplementary Fig. 1 legend for detailed design). A wooden fixing panel was placed next to the DC coupler to stabilize the position of the thin DC coupler in the camera battery slot. The flat cable connected to the DC coupler came out from the camera with the lid closed (Fig. 1a and Supplementary Fig. 1) with sealant (3 M Gel Coating GC-TCORL) on the rim of the lid to prevent corrosion. Outside the camera, the flat cable was connected to a 10 m-long DC cable. The connecting points ('connector' in Fig. 1b) were sealed with a self-fusing tape for sealing and insulation (Fig. 1a). We bound four DC cables, each of which was obtained from one camera, using a power barrel connector jack connected to an AC adaptor. The UPS station supplied power for up to 30 min in the case of a power break to prevent the interruption of photo shooting (Fig. 1a).

## Image analysis (PlantServation software)
Figure 3 summarizes the workflow of the image analysis conducted using in-house Python scripts (version 3.6.6). PlantServation demo set is available as described in Code availability.

Step 1: We selected up to four images per day that satisfied the following criteria: First, thresholding allowed only images with maximum and average pixel values greater than 80 and 10, respectively, to be analyzed. Second, the time at which an image was acquired was restricted to 10:00–14:00. These filterings reduced the variation in

brightness and sunlight direction, and the contamination of accidentally dark images.

Step 2: To adjust the position of the target plants and detect each plant location, we first detected white marbles at the corners of the plot using a DNN (ResNet), registered the image series via homography transformation, matching the marbles to reference the marbles which were determined by averaging the position of each marble, and subsequently downsized each to an image of 1152 × 864 pixels (Supplementary Fig. 2). Each of these low-resolution images was divided into nine patches of 384 × 384 pixels, and segmentation was applied to roughly assign 1 and 0 to the plant and background regions, respectively. Subsequently, we adopted both areas for overlapping parts ('AND') to fit 1152 × 864 pixels. This image was enlarged four times in length (4608 × 3456). This series of resizing and processing steps contributes to shortening the total processing time. Subsequently, we overlaid the mask images (where the object and rest were assigned to 1 and 0, respectively, to generate binary images) acquired at different time points and obtained one summed image with local peaks. Each local peak corresponds to the center of the plant from a given time point. The position of the local peaks was regarded as the position of the target plant in the image. Because the peak was distinctly separated from the background value and exhibited a distinct value, peak detection was straightforwardly performed using the peak_local_max function in the skimage feature library (https://scikit-image.org/docs/dev/api/skimage.feature.html). After peak detection, we examined whether the plants were correctly detected, and when not correctly detected, we manually identified the plant or removed the plant that had been incorrectly identified.

Step 3: Once the plant position was determined, we performed segmentation of the target plants in each 384 × 384-pixel image (Fig. 3a). Although a patch-based segmentation method using a convolutional neural network was effective for detecting fine features in a previous study[73], it was time-consuming and missed detailed features of our dataset; therefore, we used an end-to-end segmentation method with DNN using the library pytorch (https://pytorch.org/) (Supplementary Fig. 3).

The training dataset for the segmentation of the target plants was prepared as shown in Supplementary Fig. 4. First, using the annotation tool labelme, we labeled 225 images with soil background at the Swiss site consisting of plants with diverse colors, sizes, and morphologies[74]. The 225 labeled images were augmented by rotating, shifting, scaling, and changing the brightness and contrast to yield 4100 training images. We used the same 225 labeled images to generate 3400 training images for the sand background at the Swiss and Japanese sites. For this, we overlaid a labeled plant image on a randomly selected background image, from which we cut out the area of the plant. Thereafter, for each of the 7500 training images, we cut out an area of 384 × 384 pixels containing the target plant and cropped it by cutting out a circular area with a diameter of 384 pixels with the plant position as the center. This cutting procedure was effective in excluding neighboring plants. In later growth stages, some of the neighboring leaves may be included, but they are considered minor for the effect on color estimation in comparison to the fully grown focal plant. This image was used as the input for training using the best DNN architecture for our dataset. The best DNN architecture was identified as the method that yielded the most accurate segmentation results using 4100 training data points for the soil background in the Swiss site. We compared the standard U-Net[15], U-Net with pre-trained ResNet-101[75] or EfficientNet-B7[76] as a backbone, SINet[17], and DANet[54]. For all DNN architectures, we split the 4100 input data into 68%, 12%, and 20% corresponding to training, validation, and test set, respectively. DANet performed best with our dataset and was used in the analysis (Supplementary Table 2). All 7500 training images were used to further characterize the performance of the DANet model in both the soil and sand backgrounds (Supplementary Table 3). The labeling data and the time-series images

are available as Dryad datasets [https://doi.org/10.5061/dryad.1g1jwsv11] and [https://doi.org/10.5061/dryad.h70rxwdnk].

After the segmentation of the plant, post-processing was performed to remove noise. First, the output image of the DANet was subjected to Gaussian filtering at sigma = 1 (pixel). After filtering, we converted the images to black and white using thresholding. Subsequently, small distinct objects in the background of the mask images, mostly soil particles or fallen leaves of bright color, were removed by thresholding.

Step 4: We multiplied the mask image above and the raw (original) image, such that only the plant exhibited a color pixel value of more than 0, enabling the measurement of the average and median of RGB and L*a*b* values of the plant without background.

The images were processed in a Linux Ubuntu OS using eight CPUs with four cores (Intel Xeon Cache 10 MB, Intel (R) Xeon (R) CPU E5-1620 v4 @3.5 GHz), and 32 GB memory. Furthermore, we used a GPU (GeForce GTX 1080 Ti) with 11 GB graphic memory for computationally demanding tasks, including registration, augmentation, segmentation, and training using a DNN.

## Evaluation of the image analysis outcome

We prepared ground truth data (manually annotated 'correct' plant area) consisting of nine, nine, and twelve labeled single-plant images from the soil background at the Swiss site, sand background at the Swiss site, and sand background at the Japanese site, respectively. The images were selected, such that they represented variations in color, shape, and size among plants and backgrounds in all the images. Using the ground truth and outcome of the DANet, we calculated the Dice coefficient, Precision, Sensitivity, and Specificity as follows:

- Dice coefficient = 2 × area of overlap between ground truth and DANet outcome / total number of pixels in the ground truth and DANet outcome,
- Precision = tp/(tp + fp),
- Sensitivity = recall = tp/(tp + fn),
- Specificity = tn/(tn + fp),

where tp, tn, fp, and fn indicate true positives, true negatives, false positives, and false negatives, respectively.

## Leaf pigment and leaf color

To examine how appropriately the leaf color represents the anthocyanin and chlorophyll contents, we photographed the leaves of plants growing in the common garden, harvested them, and quantified anthocyanin and chlorophyll at the Swiss site, according to protocols based on the absorption spectrometry methods that detect wide molecular species of anthocyanin and chlorophyll a and b[77,78]. We included all genotypes sampled on multiple dates to cover the wide pigment accumulation levels. These plants were grown in addition to the plants for image acquisition. In yr2, one leaf per plant was subject to the study for DEN, RS7, and RS8. We had eight replicates per genotype at each of the four time points (October 31, November 20, February 14, and March 20). At each time point, we used plants that were not subject to the study at the earlier time point(s). In yr3, one leaf per plant from the remaining nine genotypes was subject to the study at each of the five time points (October 31, November 14, December 3, February 25, and April 23). We had eight replicates per genotype per time point. Due to limitations in space and the number of seeds, we used the same plants on October 31, December 3, and February 25, and on November 14 and April 23. The right and left halves of the leaf were subject to the measurement of anthocyanin and chlorophyll, respectively. The collected leaves were processed immediately or kept frozen at −80 ˚C until processed.

For the extraction of anthocyanin, each leaf tissue was ground in 1000 µL of extraction buffer (18% isopropanol and 1% HCl) and incubated at room temperature in a shaded condition for 24 h. After the

centrifugation of 10 min at 15000 $g$, the absorbance of the supernatant was measured at 535 nm (yr2) or 530 nm (yr3) and 650 nm. The relative anthocyanin content per area or weight was calculated by dividing the value ($A_{535 \text{ or } 530} - A_{650}$) by the leaf area ($mm^2$) or weight (mg). In the figures, we displayed the relative anthocyanin amount per area ($cm^2$) or weight (g) of a leaf or plant for better visibility. The leaf area was measured from the images using Fiji (Image J ver. 1.52n[79]). For the extraction of chlorophyll, each leaf tissue was immersed in a tube with 1000 µL of 100% dimethylformamide, the extraction buffer. After three days of being shaded from light in a fridge, the tube was centrifuged to spin down plant material. The absorbance of 300 µl supernatant was measured at 664 nm and 647 nm. The relative chlorophyll content was calculated as (chlorophyll a + chlorophyll b) / leaf weight (mg) where chlorophyll a and chlorophyll b were calculated as $11.65*A_{664} - 2.69*A_{647}$ and $20.81*A_{647} - 4.53*A_{664}$, respectively.

Before leaf sampling, RGB images were acquired at 150 cm above the plants using a RICOH WG-40 camera with the same setting as the time-series image acquisition (Supplementary Table 6). Before the analyses, the images were converted from the sRGB scale to the L*a*b* color space using the Python (ver 3.8.3) package grDevices. The mean pixel values were acquired from the leaf regions which had been defined as above to measure the leaf area.

Data analyses were conducted in R 4.1.0 (R Development Core Team 2021) unless otherwise stated. To examine the relationship between the relative pigment content and color information from images, we compared generalized linear models, linear models, and random forest models with the area- or weight-based anthocyanin content as the response variable and a color space (L*+a*+b*, Y + U + V, H + S + V, R + G + B) or a color index calculated from R, G, and B, i.e., Excess Red (ExR): (1.4 R − G) / (R + G + B), Green-Red Vegetation Index (GRVI): (R − G) / (R + G), or Red Green Ratio (RGR): R / G as explanatory variables[80–83]. The random forest model was developed using the *randomForest* package with default parameter settings. The details of the R packages used in this study are summarized in Supplementary Table 7. We did not include the genotype or date effect in the model to avoid overfitting and to allow genotype- and date-nonspecific estimation of the pigment. We evaluated the accuracy of the models by leave-one-out cross-validation, where one out of the 451 data points was removed from the training dataset and used for prediction in each trial and by calculating the average of the root mean square errors for 451 trials (Supplementary Figs. 27 and 28, Source Data 10). In general, the random forest model with L*a*b* performed the best for each response variable (Supplementary Fig. 6). For random forest models with L*a*b*, $R^2$ was the highest when anthocyanin per weight was a response variable (Supplementary Fig. 7). Therefore, we decided to use this model for the main analysis and examined the relative contribution of a*, b*, and L* to the anthocyanin content using the 'importance' function in *randomForest*. For results of the random forest model with L*a*b* for both anthocyanin per weight and per area, we calculated Pearson's correlation between the measured and estimated values with genotypes pooled and separated (Fig. 4 and Supplementary Figs. 8–10) and performed linear regressions on the estimated values with measured values (Fig. 4 and Supplementary Fig. 9). In addition, for anthocyanin per area, we performed a non-linear regression because of a non-linear relationship between the measured and estimated values (Supplementary Fig. 9). To examine the influence of chlorophyll on anthocyanin content, we calculated Pearson's correlation between anthocyanin per weight and chlorophyll per weight (Supplementary Fig. 11). The scripts for analyzing non-image data and generating figures are provided as Supplementary Data 1.

## Time-series plotting
To obtain a visual understanding of seasonal changes in plant traits, we plotted the plant area, L*, a*, b*, and estimated anthocyanin content in a time series. To enhance visibility, we plotted the 5-day moving averages of all plants of the same genotype per site per year using the packages *zoo*, *dplyr*, *ggplot2*, and *tidyverse*[84,85]. The number of plants per site per year is summarized in Supplementary Table 8. We removed images from one camera at the Swiss site from yr2 because the plant IDs in the images could not be confirmed at the time of the analyses. In addition, we removed anomalous data according to three criteria: [1] manual records of snow, storms, and other incidents that affect segmentation quality, [2] visual inspection of time-series plots for a*, where the cases of deviation from the norm were followed up based on the inspection of the original images to identify obstacles such as mesh and extreme cases of positioning failure, and [3] exclusion of the period of drastic value changes (outliers) in the time-series plots of the plant area. This was intended to filter out biologically unreasonable size fluctuations in the data and was performed via nearest neighbor imputation as follows: [3–1] Divide the time series into 10 blocks for each season for each site. [3–2] Mask one of the 10 blocks to be treated as a missing value. [3–3] Impute the missing values via the nearest neighbor method. [3–4] Calculate the difference between the imputed and original values. Referring to the score $z > 5$ based on the difference, we identified and removed the data from 2018/02/18 – 2018/03/12 at the Swiss site that corresponded to the period of snow, frost, and strong light (Supplementary Fig. 29, Source Data 13–18). The analyses and plots were done using Python ver. 3.9.10. The data_removed sheet in Source Data 2–7 summarizes the dates on which all data were removed from the analysis as well as the reasons for removal. Generally, the initial plant area was large at the Swiss site and small at the Japanese site (Supplementary Fig. 30). A small size increase during the season likely reflects the large starting size at the Swiss site, whereas at the Japanese site, it could be a combination of a real phenomenon with the difficulty in segmenting small plants (Supplementary Fig. 30b).

## Dimension reduction with PCA
To compare the time-series trend of the estimated anthocyanin content, we compiled data on the genotype average of the estimated anthocyanin content from three seasons from two sites into one dataset. Following previous studies analyzing time-series data, we conducted dimension reduction by PCA using the package *base* and *vegan*[86–88]. To explore the time points that likely influenced the emerging pattern, we plotted the PC1 and PC2 scores over time using the R package *tidyverse*. To examine whether the trend was consistent between the two sites, we also conducted PCAs on the dataset in which the two sites were separated.

In addition, we investigated how well the average genotype represented the variation within the genotype. Accordingly, we split the dataset into sites and years, as the individuals were not consistent across sites or years. For each dataset, we performed Bayesian PCA using the *pcaMethods* package. This method complements missing values suitably when they constitute up to approximately 15% of the data, which is the case in our datasets (Supplementary Figs. 23 and 24).

## Environmental data collection
To record environmental conditions, we set up weather stations at each site for precipitation, radiation, and air temperature. Precipitation data contained both rain and snow. The WS GP2 weather station (Delta-T Devices) was installed at the Swiss site, whereas at the Japanese site, different parameters were collected using devices from different suppliers (Supplementary Table 9). The data were collected at 10 and 5 min intervals at the Japanese and Swiss sites, respectively.

Because of interruptions in data collection, precipitation and radiation data in the Swiss site were substituted by data from a nearby weather station of the Swiss Federal Office of Meteorology and Climatology (Source MeteoSwiss) in Fluntern. Missing values in air temperature data in the Swiss site were estimated using the *imputeTS* package.

## Relationship between anthocyanin and environmental factors

To analyze how the estimated anthocyanin content is associated with environmental factors, we included air temperature, radiation, and precipitation as the key environmental factors in the analysis based on previous studies, field observations, and laboratory experiments[23,24,48,56] (Supplementary Fig. 31). The threshold temperatures of 1, 4, 7, 10, and 13 °C were set for the calculation of coldness from air temperature to cover the range adopted or identified as critical in previous studies on cold treatment in *A. thaliana*[24,89–92] (Fig. 6a). Considering the response time of plants to environmental cues, we determined for each environmental factor "reference windows," i.e., durations for accumulating signals for anthocyanin content, and "lags," i.e., durations between signal accumulation and anthocyanin content of a given day (Fig. 6a). The ranges of the reference windows and lags were 1–14 days and 0–14 days, respectively. To model for the period when the plants were growing in the field, anthocyanin data for the first 28 days (= max. window 14 days + max. lag 14 days) for each yr per site were excluded from the analysis. We split the dataset into Swiss and Japanese sites to account for the differences in climate and the trend of fluctuation in the estimated anthocyanin content. For each genotype, we fitted linear regression models with coldness, radiation, and precipitation as explanatory variables after standardization and estimated anthocyanin content as a response variable using one of the 46,305,000 parameter combinations (5 temperature thresholds × $14^3$ windows × $15^3$ lags). We selected the model with the smallest Akaike information criteria as the best model and used it to examine the association between environmental factors and estimated anthocyanin content. These calculations were performed using R. 4.2.0. As the assumptions of the linear model were not always met, we determined the significance of the explanatory variables by calculating confidence intervals with the bias-corrected and accelerated bootstrap in the *boot* package. The relative contributions of the explanatory variables were extracted using the *relaimpo* package. In addition, we generated time-series plots of the estimated anthocyanin content and environmental factors with the best parameters for the model plant *A. thaliana* using the *ggplot2* and *reshape2* packages.

## Reporting summary

Further information on research design is available in the Nature Portfolio Reporting Summary linked to this article.

## Data availability

Source data are provided with this paper. The time-series image data for the Swiss site generated in this study have been deposited in a Dryad repository [https://doi.org/10.5061/dryad.1g1jwsv11][93]. The time-series image data for Japanese site generated in this study as well as the labeling data for image analysis used in this study are available in a Dryad repository [https://doi.org/10.5061/dryad.h70rxwdnk][94]. Note that each Dryad datasets is large, nearly 250 GB. Source data are provided with this paper.

## Code availability

The scripts used for the analyzes of non-image data are provided as Supplementary Data 1. The PlantServation demo set (ca. 600 MB) including scripts and demo data for PlantServation software is available at Zenodo [https://zenodo.org/record/7321725][95] accessible via Dryad repository [https://doi.org/10.5061/dryad.h70rxwdnk][94].

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

## Acknowledgements

The authors thank Dennis Bailer, Richard Baxter, Gioele Bello, Silas Braun, Silvana Capaul, Mizuki Sato, Marcel Freund, Matthias Furler, Anne Graf, Nangsa Karutshang, Lucas Mohn, Aki Morishima, Atsushi J. Nagano, Kenji Ogawa, Elin Rütimann, Yasuhiro Sato, Reinhold Stockenhuber, Karen Thomsen, Misako Yamazaki, Mayumi Yorino, Center for Ecological Research of Kyoto University, and Kudoh Lab at Kyoto University. The services have been provided by MeteoSwiss, the Swiss Federal Office of Meteorology and Climatology. This study was funded by Core Research for Evolutionary Science and Technology grant number JPMJCR16O3 to K.K.S., Y.S., J. Sese and T.K., JPMJCR15O1 to H.K., Swiss National Science Foundation grant 310030_212551, 31003A_182318 and CRSII5_183578 to K.K.S. and 310030_212674 to R.S.-I., Global Affairs of the University of Zurich to K.K.S., Kakenhi 22H02316 to K.K.S., 17H06990 to T.T., 21H04977 to H.K., and 22H05179 to K.K.S. and J. Sun, JSPS International Leading Research grant 22K21352 to K.K.S. and H.K., University Research Priority Program in Evolution in Action, the University of Zurich, to K.K.S. and R.S.-I., and Joint Usage/ Research Grant of Center for Ecological Research (2017jurc-cer05, 2018jurc-cer09, and 2019jurc-cer02), Kyoto University, to R.S.-I.

## Author contributions

These authors contributed equally: R.A., T.G., and T.T. R.A., R.S.-I. designed the experiment, R.A., T.T., J. Sugisaka, and R.S.-I. conducted the experiment, R.A., T.G., T.T., M.H., K.K., and J. Sun analyzed the data, R.A., T.G., T.T., and K.K.S. wrote the manuscript, J.A., N.K., A.T., H.K., T.K., and J. Sese provided technical support, and Y.S., J. Sese, N.K., K.K.S. conceived the study. All authors contributed to revising the manuscript and approved the final version.

## Competing interests

The authors declare no competing interests.

## Additional information

[1]Department of Evolutionary Biology and Environmental Studies, University of Zurich, Winterthurerstrasse 190, CH-8057 Zurich, Switzerland. [2]Research and Development Division, LPIXEL Inc., Chiyoda-ku, Tokyo 100–0004, Japan. [3]Kihara Institute for Biological Research (KIBR), Yokohama City University, 641-12 Maioka, Totsuka-ward, Yokohama 244-0813, Japan. [4]Division of Biological Science, Graduate School of Science and Technology, Nara Institute of Science and Technology (NAIST), 8916-5 Takayama-Cho, Ikoma, Nara 630-0192, Japan. [5]Center for Ecological Research, Kyoto University, Hirano 2-509-3, Otsu 520-2113, Japan. [6]Department of Biological Sciences, Graduate School of Science, The University of Tokyo, 7-3-1 Hongo, Bunkyo-ku, Tokyo 113-0033, Japan. [7]Research Center for Agricultural Information Technology, National Agriculture and Food Research Organization, 3-1-1 Kannondai, Tsukuba, Ibaraki 305-8517, Japan. [8]Department of Electric and Computer Engineering, Kanazawa University, Kakuma, Kanazawa 920-1192, Japan. [9]Functional Genomics Center Zurich, Winterthurerstrasse 190, CH-8057 Zurich, Switzerland. [10]Sugadaira Research Station, Mountain Science Center, University of Tsukuba, 1278-294 Sugadaira-kogen, Ueda 386-2204, Japan. [11]Artificial Intelligence Research Center, AIST, 2-3-26 Aomi, Koto-ku, Tokyo 135-0064, Japan. [12]Humanome Lab, Inc., L-HUB 3F, 1-4, Shumomiyabi-cho, Shinjuku, Tokyo 162-0822, Japan. [13]AIST-Tokyo Tech RWBC-OIL, 2-12-1 O-okayama, Meguro-ku, Tokyo 152-8550, Japan. ✉e-mail: rie.inatsugi@ieu.uzh.ch; kentaro.shimizu@uzh.ch

