## [Peer Review File · Nature Communications]

Reviewers' Comments:

Reviewer #1:

Remarks to the Author:

This ms is well written and addresses a number of interesting topics: the application of machine learning to capturing plant phenotypes in relation to environmental variables, the variation in anthocyanins as a stress indicator across seasons and conditions, and the accumulation of anthocyanins in polyploids and their diploid progenitors. Each of these is a potentially important topic, and there is the potential to pull all of this information together in an impactful way. However, I do not think that the current ms completely accomplishes this goal; rather, the pieces seem somewhat separate. I could envision this as three separate papers, each valuable in its own right, but I could also envision this as a stronger paper that thoroughly integrates the parts. For example, the paper really emphasizes the machine learning aspect, with the anthocyanins and the biology that those comparisons represent as a case study. That would be fine, but it should be written as such: clearly indicate that the anthocyanin aspect is a case study. As is, the paper switches suddenly from machine learning to anthocyanins to polyploidy. Alternatively, the paper could really focus on either the anthocyanins or the polyploidy as the lead but indicate the need for imaging pipelines and then introduce the PlantServation aspect. Of course, the current approach could also be fine, but it requires more integration, as noted above.

In addition, although I think that the pipeline is well described and an important application, it is unclear that this same approach would work for others (but perhaps the details of hardware and code would allow replicability – I would hope so). And is it so much different from what many other phenotyping projects in agriculture or plant biology have used? Certainly, some of the papers in the June-July 2020 special issue of *Applications in Plant Biology*, focused on machine learning in plant biology, address similar questions and provide descriptions of phenotyping, segmentation, etc. This makes me wonder about the novelty of this specific application (although a lack of novelty does not reduce its value for the specific applications addressed here).

I also have some concerns about relating leaf color to anthocyanin content. The authors inferred anthocyanin content, related to color, by estimating content and then dividing it by leaf area. However, many factors other than leaf area could be involved in an accurate estimate of anthocyanin content, and this is especially true when content is inferred from color. For example, leaf thickness and cell number are both important when actually putting content on a per-unit basis. Dry weight or fresh weight would have been better than area; also, because anthocyanins are typically located in the cell vacuole, cell number would be a good metric, but a per-cell estimate would be much more involved (requiring a relationship between cell number and leaf volume). I am therefore a bit concerned about putting the content on a per-area basis. When relating to color, I also have some concerns. Anthocyanins exist in leaves along with many other pigments, especially chlorophyll. That means that a given anthocyanin content might appear as pink, red, or green leaves, depending on the amount of chlorophyll also present in the leaves, which could vary depending on season, stress, etc., but is unknown. The impact of variation in other pigments and cuticle is also unknown. Thus, I am not sure that anthocyanin content is actually being measured; rather, the color is recorded, but the relationship to actual content seems unclear. A better metric would be color as an indicator of stress, induced experimentally. It would also be interesting to know if the same or different compounds are being produced, both among genotypes and species and over time. This might perhaps be a more interesting question than content.

I also felt that the treatment of anthocyanins in the polyploids and diploid parents was superficial. There is a large literature on anthocyanins and other flavonoids in polyploids and their progenitors, and this is not addressed. The current paper, though, with these very approximate estimates of content, does not actually contribute much to the very interesting issues raised over the past decades about the diversity of compounds that could be produced in an allopolyploid in particular, given the combination of pathways of its parents.

In addition, I think that all data and code should be publicly available, with nothing held back to be obtained from the authors upon request.

In sum, I find all aspects of the ms interesting, but I also think that each part would require work to meet the standards of the Nature family of journals. I also think that the anthocyanin portions of the paper require much more substantial work to be useful as verifiable and solid markers of stress and/or polyploidy.

Reviewer #2:

Remarks to the Author:

This study reports on time-series planting phenotyping for detecting seasonal fluctuation in anthocyanin content in diploid and polyploid Arabidopsis. The authors detected automated hardware and software systems to collect images in the fields for this experimental study. And a large dataset was generated to cover three years and two sites in Japan and Switzerland. The time-series monitoring of anthocyanin content for the entire growing season in response to environmental changes is novel. It is important for answering biological questions. I have little knowledge about the biological processes and my concerns are mainly focused on the estimation of anthocyanin content from color information.

1.Figs. 4 & S6: the anthocyanin content (Anth) is usually expressed in $\mu\text{g}/\text{cm}^2$ and ranges from 0 to 40. The expression in this study is inappropriate. The term relative anthocyanin content per mm^2 is confusing. If it is expressed per leaf area, then it is not a relative value but an absolute value. Moreover, I suggest to use cm^2 instead of mm^2 , so that the digital numbers do not have to carry too many digits (E-4).

2.The data points in Fig. 4 were concentrated at the lower end and contributed more to the correlation coefficient than the points at the higher end. The r value might be a bit misleading and could not reflect the diversion of data points from the 1:1 line at the higher end. There should be a better way to measure the goodness of fit in this case. There were too many data points representing green leaves. Is it possible to build a random forest model for purple leaves specifically?

3.Line 239-240: underestimation is serious for high values (>0.015) where the predicted values were all below the 1:1 line. This insensitivity to high Anth may cause problems for the leaves with high Anth. This may be a problem for purple leaves in the field. In the time series data of Fig. 5, the predicted values were all lower than 0.0125 for the time-series phenotyping over three years and two sites. Was this true or caused by the color saturation at high Anth values? If it was caused color saturation, the predicted Anth for purple leaves might not be convincing.

4.Fig. S14: The LOO CV accuracy is low with a R^2 of 0.45. Why was the accuracy in Fig. S6 much higher? Is it hard to believe such a model could work well for the time-series phenotyping project.

5.Table S4: If the explanatory variables only included $R+G+B$ and $L^*+a^*+b^*$ for Anth estimation, more variables could be extracted from the RGB images. There are many studies on the extraction of color indices generated from two or three channels in the literature of remote sensing and plant phenotyping. Advanced methods for pigment estimation could be helpful for fixing the color saturation problem. This is the key to improved phenotyping for tracking the temporal variation in Anth in response to complex environmental conditions.

Reviewer #3:

Remarks to the Author:

This study highlights an inexpensive and suitable approach to phenotyping plant populations in natura; and uses it to address the consequences of hybrid allopolyploidy with an elegant design. The choice to address anthocyanin as a phenotype known to show high environmental plasticity is of great interest to address how the combination of two progenitor species in a third species interacts with environmental variation.

The study makes the best of Arabidopsis-relatives, including *A. thaliana* to validate findings as well as taxa of the polyploid complex in suitably replicated sites. The use of experimentally

resynthesized polyploids and their comparison with naturally established ones is of particular fundamental interest.

It matches papers typically published in Nature Communications, offering a detailed description of a solid amount of data and analyses that may inspire follow-up studies relying on this methodology. It also brings our understanding of polyploid systems further, and will hopefully foster further use of experimental allopolyploids in an ecological context.

I noticed only one minor issue that I was unable to solve through the rich documentation: L. 515 about the experimental production of synthetic allopolyploids "RS7 was automatically polyploidized from a hybrid...", you mean "spontaneously" (i.e. without induction by colchicine). By the way, it may be good to spell their details out (are they S0?, aso)

REVIEWER COMMENTS

Reviewer #1 (Remarks to the Author):

This ms is well written and addresses a number of interesting topics: the application
of machine learning to capturing plant phenotypes in relation to environmental
variables, the variation in anthocyanins as a stress indicator across seasons and
conditions, and the accumulation of anthocyanins in polyploids and their diploid
progenitors. Each of these is a potentially important topic, and there is the potential
to pull all of this information together in an impactful way. However, I do not think
that the current ms completely accomplishes this goal; rather, the pieces seem
somewhat separate. I could envision this as three separate papers, each valuable in
its own right, but I could also envision this as a stronger paper that thoroughly
integrates the parts. For example, the paper really emphasizes the machine learning
aspect, with the anthocyanins and the biology that those comparisons represent as a
case study. That would be fine, but it should be written as such: clearly indicate that
the anthocyanin aspect is a case study. As is, the paper switches suddenly from
machine learning to anthocyanins to polyploidy. Alternatively, the paper could really
focus on either the anthocyanins or the polyploidy as the lead but indicate the need
for imaging pipelines and then introduce the PlantServation aspect. Of course, the
current approach could also be fine, but it requires more integration, as noted above.

Thank you for your comment on the presentation of the manuscript. In the revised manuscript
we formulate such that we developed PlantServation (phenotyping method using machine
learning) and applied it to quantifying anthocyanin in diploids and poplyploids as a case
study (Abstract line 45, Introduction lines 141–143).

In addition, although I think that the pipeline is well described and an important
application, it is unclear that this same approach would work for others (but perhaps
the details of hardware and code would allow replicability – I would hope so). And is
it so much different from what many other phenotyping projects in agriculture or
plant biology have used? Certainly, some of the papers in the June-July 2020 special
issue of Applications in Plant Biology, focused on machine learning in plant biology,
address similar questions and provide descriptions of phenotyping, segmentation,
etc. This makes me wonder about the novelty of this specific application (although a
lack of novelty does not reduce its value for the specific applications addressed here).

We now provide the demo dataset, script, and a manual (README) in crest_demo_cpu.zip
(ca. 600 MB) so that one can check the reproducibility of the pipeline. It is available as a part
of Supplementary Information. It is also available as a part of a dataset deposited in Dryad,
however, please note that **the entire dataset (nearly 250 GB) will be downloaded in one go**
when clicking the link for Dataset 2/2 in Data availability section of the revised manuscript.

As the reviewer points out, there have been other phenotyping methods using machine
learning in plant research. The characteristic of our phenotyping method lies in the

integration of strengths in that it is inexpensive, robust, and able to handle a large number of
noisy and high-resolution images from the field by efficiently utilizing DNNs. As such, it
overcomes challenges listed in the first two paragraphs of Introduction and enables the
analysis of time-series images of small plants in the field. This, in turn, paves a way to
address biological questions which have been difficult otherwise. We modified the final part
of the first and second paragraphs of Introduction to highlight the characteristics of our
method (lines 81–83, 101–104). In the revised manuscript, the former reads ‘*Overcoming all*
*these challenges and analyzing time-series images of different species in different*
*environments further our understanding of the growth and environmental responses of*
*plants.*’. The latter now reads ‘*The application of DNN to high-resolution image analysis of*
*plants in the field while overcoming the challenges described in this and the previous*
*paragraphs enables the identification of diverse biological questions, including ecology and*
*evolution, with pigment accumulation in allopolyploids and their progenitors being one an*
*example.*’ In addition, we articulated the challenges in field phenotyping in the middle of the
same paragraph citing Champ et al. 2020 (Appl Plant Sci 8: 1-20) (line 93, reference #15).

I also have some concerns about relating leaf color to anthocyanin content. The
authors inferred anthocyanin content, related to color, by estimating content and
then dividing it by leaf area. However, many factors other than leaf area could be
involved in an accurate estimate of anthocyanin content, and this is especially true
when content is inferred from color. For example, leaf thickness and cell number are
both important when actually putting content on a per-unit basis. Dry weight or
fresh weight would have been better than area; also, because anthocyanins are
typically located in the cell vacuole, cell number would be a good metric, but a per-
cell estimate would be much more involved (requiring a relationship between cell
number and leaf volume). I am therefore a bit concerned about putting the content
on a per-area basis. When relating to color, I also have some concerns. Anthocyanins
exist in leaves along with many other pigments, especially chlorophyll. That means
that a given anthocyanin content might appear as pink, red, or green leaves,
depending on the amount of chlorophyll also present in the leaves, which could vary
depending on season, stress, etc., but is unknown. The impact of variation in other
pigments and cuticle is also unknown. Thus, I am not sure that anthocyanin content
is actually being measured; rather, the color is recorded, but the relationship to
actual content seems unclear. A better metric would be color as an indicator of
stress, induced experimentally. It would also be interesting to know if the same or
different compounds are being produced, both among genotypes and species and
over time. This might perhaps be a more interesting question than content.

Thank you for sharing your insights.

Upon your suggestion, we utilized available data and estimated not only anthocyanin per
area, but also anthocyanin per fresh weight from color information. The results of leave-one-
out-cross-validation indicate that the random forest model with $L^*a^*b^*$ with anthocyanin per
weight show higher correlation coefficient for measured and predicted values ($R^2 = 0.64$)
compared with anthocyanin per area (Supplementary Fig. 7, 27, and 28). The linearity in the
fitting plot, especially for the anthocyanin value range 0-50 (Fig. 4), shows the validity of

anthocyanin per weight. Unlike previous study on *Arabidopsis thaliana* in the laboratory
which found a clear opposite trend between anthocyanin and chlorophyll along color gradient
(Faragó et al. 2018 *Frontiers in Plant Science* 9:1–12, line 264, reference #48), anthocyanin
and chlorophyll were only weakly associated in our dataset (Supplementary Fig. 11). The
weak association despite a wide range of chlorophyll data points suggests that the influence
of chlorophyll alone on anthocyanin is not great in our dataset possibly because other factors
such as light and leaf wax are also influential in outdoor condition. This weak association
suggests that the validity of our anthocyanin data would not be affected by chlorophyll.
Given these, in the revised version of the manuscript, we adopted anthocyanin per weight.
With anthocyanin per weight, we can capture the seasonal fluctuation of anthocyanin which
is the main goal of the study. The higher accuracy of anthocyanin per weight in comparison
to anthocyanin per area could be attributed to the size of the cells with anthocyanin: if the
cells in layers beneath the surface layer are larger, they may accumulate more anthocyanin,
leading to the anthocyanin content better estimated when weight based. Please note that most
of our conclusions of the downstream analyses were not affected by changing the estimation
from anthocyanin per area (Supplementary Fig. 12, 16, 18, 19, 21, 24, and 26) to that per
weight (Fig. 5–7, Supplementary Fig. 17, 20, 23, and 25). We appreciate your precious
suggestion to improve our estimation of the anthocyanin content.

In addition to the anthocyanin contents, we performed PCA on a* to examine the pattern of
seasonal fluctuation of color (Supplementary Fig. 22). The result of a* resembled those of the
anthocyanin contents (Fig. 7 and Supplementary Fig. 19), suggesting that the evolutionary
relationship implied by the similarity among the plants in this study is not an artifact of the
anthocyanin estimation model.

It is beyond the scope of this study to experimentally manipulate stress or quantify
compounds, however, these would provide interesting further insights into the environmental
response of the genotypes and species over time. The observed color change in indoor *A.*
*halleri* in response to light and temperature suggests that such experiments are promising in
studying compounds (Supplementary Fig. 31).

I also felt that the treatment of anthocyanins in the polyploids and diploid parents
was superficial. There is a large literature on anthocyanins and other flavonoids in
polyploids and their progenitors, and this is not addressed. The current paper,
though, with these very approximate estimates of content, does not actually
contribute much to the very interesting issues raised over the past decades about the
diversity of compounds that could be produced in an allopolyploid in particular,
given the combination of pathways of its parents.

Thank you for raising this point. As described in the Results (related to Fig. 6) and
Discussion, our focus was on characterizing environmental response patterns among species
and genotypes using anthocyanin as an indicator, rather than characterizing different pigment
compounds and their pathways. Diversity in the molecular details and pathways of pigments,
as well as diversity in the sensing mechanisms for environmental cues, may account for the
observed diversity in anthocyanin fluctuation. These are beyond the scope of this study but
highlighted by our results as future challenges (Discussion, lines 457–461).

Although this study was set up to address the diversity in seasonal fluctuation patterns of leaf
anthocyanin from time-series data in the field, there is a common finding in this study and in
the literatures on the diversity of pigments and colors of flowers in allopolyploids under
controlled condition. Thus, we revised the Discussion by incorporating McCarthy et al. 2017
(Am J Bot 104: 92–101) and McCarthy et al. 2015 (Ann Bot 115: 1117–1131) (references
#62 and 63, lines 479–484). The corresponding part reads:

*The similarity of synthetic allopolyploids to a subset of natural counterparts that exhibit*
*variation in a trait is consistent with the findings on the pigments (cyanidin, quercetin, and*
*kaempferol) and color in Nicotiana flowers under controlled conditions^{62,63}. Our time-series*
*data suggest that synthetic polyploids can recapitulate the polyploid speciation that is*
*observable in the fluctuation of anthocyanin content and leaf color in outdoor conditions.*

In addition, I think that all data and code should be publicly available, with nothing
held back to be obtained from the authors upon request.

We deposit the data and scripts from this study in Supplementary Information or in the public
repository Dryad. The details of the items in Dryad are stated in the Data availability and
Code availability sections of the revised manuscript.

In sum, I find all aspects of the ms interesting, but I also think that each part would
require work to meet the standards of the Nature family of journals. I also think that
the anthocyanin portions of the paper require much more substantial work to be
useful as verifiable and solid markers of stress and/or polyploidy.

We hope the revised manuscript provides a better framework and that the adoption of
anthocyanin per weight enhances the quality of the study to meet the standards of the journal.

Reviewer #2 (Remarks to the Author):

This study reports on time-series planting phenotyping for detecting seasonal
fluctuation in anthocyanin content in diploid and polyploid Arabidopsis. The authors
detected automated hardware and software systems to collect images in the fields
for this experimental study. And a large dataset was generated to cover three years
and two sites in Japan and Switzerland. The time-series monitoring of anthocyanin
content for the entire growing season in response to environmental changes is novel.
It is important for answering biological questions. I have little knowledge about the
biological processes and my concerns are mainly focused on the estimation of
anthocyanin content from color information.

1.Figs. 4 & S6: the anthocyanin content (Anth) is usually expressed in ug/cm² and
ranges from 0 to 40. The expression in this study is inappropriate. The term relative
anthocyanin content per mm² is confusing. If it is expressed per leaf area, then it is
not a relative value but an absolute value. Moreover, I suggest to use cm² instead of
mm², so that the digital numbers do not have to carry too many digits (E-4).

According to the comment by the Reviewer 1, we thoroughly revised this point.

In the revised version of the manuscript, we changed the unit to relative amount per leaf
weight which improved the fitting of the values, and we used g as a unit in figures to make
the range of the digits more visible. Please, refer to our comments to Reviewer 1 for the
details of the adoption of anthocyanin per weight instead of anthocyanin per area.

As to anthocyanin per area, we adopted cm^2 instead of mm^2 in the figures to improve
visibility. Besides, we display the amount for the entire sampled leaf area because the original
version showed the amount for a half of the sampled leaf area by mistake.

In addition, as we measured the anthocyanin content by absorption spectrophotometry, the
accurate conversion of absorbance to weight (μg) is not possible due to the complexity of
molecular species of anthocyanin. We retained the term relative following the description in a
previous study (Neff and Chory 1998, Plant Physiology 118:27-36,
<https://doi.org/10.1104/pp.118.1.27>.)

2.The data points in Fig. 4 were concentrated at the lower end and contributed more
to the correlation coefficient than the points at the higher end. The r value might be
a bit misleading and could not reflect the diversion of data points from the 1:1 line at
the higher end. There should be a better way to measure the goodness of fit in this
case. There were too many data points representing green leaves. Is it possible to
build a random forest model for purple leaves specifically?

The adoption of anthocyanin per weight instead of per area resolved the diversion of the data
points from the = 1:1 line and the concentration of data points corresponding to green leaves
to a great extent (Fig. 4).

It is technically possible to split the dataset at an arbitrary color threshold and run a random
forest model respectively. However, few data points showing large variation for red leaves
are not apt for accurate estimation.

3.Line 239-240: underestimation is serious for high values (>0.015) where the
predicted values were all below the 1:1 line. This insensitivity to high Anth may cause
problems for the leaves with high Anth. This may be a problem for purple leaves in
the field. In the time series data of Fig. 5, the predicted values were all lower than
0.0125 for the time-series phenotyping over three years and two sites. Was this true
or caused by the color saturation at high Anth values? If it was caused color
saturation, the predicted Anth for purple leaves might not be convincing.

As explained above, the diversion of the data points from the = 1:1 line generally improved
by adopting anthocyanin per weight.

The deviation is conspicuous when the value of the estimated anthocyanin content is >50 ,
which is of extremely red leaves we included to avoid extrapolation in the time-series data
analysis (Fig. 4). In our time-series data, the values of the estimated anthocyanin contents are

<50 (Fig. 5). Similarly, the values of L^* , a^* , and b^* in our time-series are within the range of
those in the dataset for pigment measurement (L^* : 11.99982-193.9581, a^* : 106.5-156.0156,
b^* : 91.625-178.1094). Given these, we consider that the influence of the deviation from the =
1:1 line on the analysis of the time-series trend is limited and that we can capture the essential
trends of the seasonal fluctuation of the anthocyanin content.

Furthermore, the deviation is in such a manner that the anthocyanin content is underestimated
for purple leaves. This leads to a conservative evaluation of the difference in the anthocyanin
content between plants with large and small anthocyanin contents. As such, we could
interpret the detected difference in anthocyanin content between species and genotypes with
confidence.

4.Fig. S14: The LOO CV accuracy is low with a R^2 of 0.45. Why was the accuracy in Fig.
S6 much higher? Is it hard to believe such a model could work well for the time-
series phenotyping project.

With anthocyanin content per weight, R^2 from the LOO CV is 0.64. The difference in the
accuracy between Fig. S14 (corresponding to Supplementary Fig. 28 in the revised
manuscript, with corrected R^2 value 0.44 as the original value was calculated for a subset of
the data by mistake) and Fig. S6 (corresponding to Supplementary Fig. 10 in the revised
manuscript) is due to that the former shows the result of cross validation whereas the latter
shows that of fitting. With LOO CV we divided the data into subsets and repeated training
and validation to obtain the result that shows how good the model is at predicting the
anthocyanin content. With fitting we used all the data to determine the decision tree
parameters of the model.

Even if each single estimation contains a certain noise, we consider that we can capture the
essential trends of the seasonal fluctuation of the anthocyanin content with the current
accuracy based on the average of multiple individual plants and adoption of a moving
average which reduces noise (e.g., Smith 1997, The Scientist and Engineer's Guide to Digital
Signal Processing Chapter 15, P279; Warner 2016, Optimizing the Display and Interpretation
of Data, Chapter 3, P56) (references #82 and 83). In addition, the trend detected by using the
model corresponds to the known phenomenon of *Arabidopsis* to turn reddish under low
temperatures in winter, suggesting the model to be biologically reasonable.

5.Table S4: If the explanatory variables only included $R+G+B$ and $L^*+a^*+b^*$ for Anth
estimation, more variables could be extracted from the RGB images. There are many
studies on the extraction of color indices generated from two or three channels in
the literature of remote sensing and plant phenotyping. Advanced methods for
pigment estimation could be helpful for fixing the color saturation problem. This is
the key to improved phenotyping for tracking the temporal variation in Anth in
response to complex environmental conditions.

Thank you for your suggestion. Referring to the literature of remote sensing and plant
phenotyping, we compared RMSE and R^2 of LOO CV for different pigment estimation
methods that were applicable to our data of leaf pigments measured at specific wave lengths
with absorption spectrophotometry. Of the examined methods, i.e., $R+G+B$, $L^*+a^*+b^*$,
$Y+U+V$, $H+S+V$, Excess Red, Green Red Vegetation Index, and Red Green Ratio, it turned

out that a random forest model with $L^*+a^*+b^*$ and the anthocyanin content per weight was
the most accurate (Supplementary Fig. 6–7). Therefore, we adopted this model throughout
the revised manuscript.

Reviewer #3 (Remarks to the Author):

This study highlights an inexpensive and suitable approach to phenotyping plant
populations in natura; and uses it to address the consequences of hybrid
allopolyploidy with an elegant design. The choice to address anthocyanin as a
phenotype known to show high environmental plasticity is of great interest to
address how the combination of two progenitor species in a third species interacts
with environmental variation.

The study makes the best of Arabidopsis-relatives, including *A. thaliana* to validate
findings as well as taxa of the polyploid complex in suitably replicated sites. The use
of experimentally resynthesized polyploids and their comparison with naturally
established ones is of particular fundamental interest.

It matches papers typically published in Nature Communications, offering a detailed
description of a solid amount of data and analyses that may inspire follow-up studies
relying on this methodology. It also brings our understanding of polyploid systems
further, and will hopefully foster further use of experimental allopolyploids in an
ecological context.

I noticed only one minor issue that I was unable to solve through the rich
documentation: L. 515 about the experimental production of synthetic allopolyploids
"RS7 was automatically polyploidized from a hybrid...", you mean "spontaneously"
(i.e. without induction by colchicine). By the way, it may be good to spell their details
out (are they S0?, aso)

Thank you for your suggestion. In the revised manuscript, we rephrased the polyploidization
process as suggested (line 552) and provided the generation and other information about the
synthetic allopolyploids (Methods, lines 554–557).

Reviewers' Comments:

Reviewer #1:

Remarks to the Author:

I appreciate the authors' efforts to revise the ms following the suggestions of the reviewers, where appropriate. I think the revisions have made for a much stronger ms, and I have no further substantive suggestions. I noted a few small editorial issues, but I assume that they will be corrected by the copy editor. Examples are: (1) line 463: 'in fields' should be 'in the field' and (2) line 815: 'was' should be 'were' ('data' is plural).

Reviewer #2:

Remarks to the Author:

I am grateful that the authors had made substantial revisions to improve the manuscript. The estimation of anthocyanin content from color information has been improved remarkably by changing the unit from area basis to fresh weight basis. I have no more comments.

**REVIEWERS' COMMENTS**

Reviewer #1 (Remarks to the Author):

I appreciate the authors' efforts to revise the ms following the suggestions of the
reviewers, where appropriate. I think the revisions have made for a much stronger ms,
and I have no further substantive suggestions. I noted a few small editorial issues, but I
assume that they will be corrected by the copy editor. Examples are: (1) line 463: 'in
fields' should be 'in the field' and (2) line 815: 'was' should be 'were' ('data' is plural).

Thank you for your thorough review and comment.

We corrected the expressions pointed out in (1) and (2) and reviewed and corrected other
editorial issues in the manuscript.

We highly appreciate your suggestions to improve the manuscript.

Reviewer #2 (Remarks to the Author):

I am grateful that the authros had made substantial revisions to improve the mansucrypt.
The estimation of anthocyanin content from color information has been improved
remarkably by changing the unit from area basis to fresh weight basis. I have no more
comments.

Thank you for your comment. We highly appreciate your suggestions to improve the
manuscript.